# A Comprehensive Framework for Stochastic Calibration and Sensitivity Analysis of Large-Scale Groundwater Models

Andrea Manzoni[1], Giovanni Michele Porta[1], Laura Guadagnini[1], Alberto Guadagnini[1], Monica Riva[1]

[1]Dipartimento di Ingegneria Civile e Ambientale (DICA), Politecnico di Milano, Milano, 20133, Italy

*Correspondence to*: Monica Riva (monica.riva@polimi.it)

**Abstract.**

We introduce a comprehensive and robust theoretical framework and operational workflow that can be employed to enhance our understanding, modeling and management capability of complex heterogeneous large-scale groundwater systems. Our framework encapsulates key components such as the three-dimensional nature of groundwater flows, river-aquifer interactions,

probabilistic reconstruction of three-dimensional spatial distributions of geomaterials and associated properties across the subsurface, multi-objective optimization for model parameter estimation through stochastic calibration, and informed global sensitivity analysis. By integrating these components, we effectively consider the inherent uncertainty associated with subsurface system characterizations as well as their interactions with surface water bodies. The approach enables us to identify parameters impacting diverse system responses. By employing a coevolutionary optimization algorithm, we ensure efficient

model parameterization, facilitating simultaneous and informed optimization of the defined objective functions. Additionally, estimation of parameter uncertainty naturally leads to quantification of uncertainty in system responses. The methodology is designed to increase our knowledge of the dynamics of large-scale groundwater systems. It also has the potential to guide future data acquisition campaigns through the informed global sensitivity analysis. We demonstrate the effectiveness of our proposed methodology by applying it to the largest groundwater system in Italy. We address the challenges posed by the

characterization of the heterogeneous spatial distribution of subsurface attributes across large-scale three-dimensional domains upon incorporating a recent probabilistic hydrogeological reconstruction specific to the study case. The system considered faces multiple challenges, including groundwater contamination, sea water intrusion, and water scarcity. Our study offers a promising modeling strategy applicable to large-scale subsurface systems and valuable insights into groundwater flow patterns that can then inform effective system management.

**Keywords:** large-scale groundwater modeling, multi-objective optimization function, global sensitivity analysis, coevolutionary algorithm, uncertainty quantification.

# 1 Introduction

Large-scale groundwater flow models have been developed in recent years (e.g., Maxwell et al., 2015; Naz et al., 2023) in response to growing interest in understanding potential impacts of climate and anthropogenic drivers on water systems as well as in assessing large-scale patterns and processes affecting water security. This progress has been facilitated by an increased availability of data and computational capabilities (e.g., Zhou and Li, 2011; Amanambu et al., 2020 and references therein). Building such large-scale models often requires to consider important simplifications. In some cases, constant properties are assumed along the vertical direction (e.g., Maxwell et al., 2015; Shrestha et al., 2014; Soltani et al., 2022) without taking into account the three-dimensional nature of the spatial heterogeneity of the subsurface system. In addition, parametrization of these models does not rely on rigorous model calibration against data that are, in turn, typically scarce. Instead, parameter values are typically inferred from literature information (Naz et al., 2023; Maxwell et al., 2015), thus possibly introducing large margins of uncertainty that are seldom quantifiable. The work of Manzoni et al. (2023) addresses these challenges by proposing a machine-learning-based methodology for delineating the spatial distribution of geomaterials across large-scale three-dimensional subsurface systems. These authors showcase their approach upon focusing on the Po River Basin in northern Italy. Their work provides a comprehensive dataset comprising lithostratigraphic data from various sources and offers a robust framework for quantifying prediction uncertainty at each spatial location within the reconstructed domain. Hence, our study rests on the findings of Manzoni et al. (2023). In these sense, the latter serve as more than simply a dataset but as a critical component upon which we build our groundwater flow model calibration. Although the literature includes examples of large/national-scale models covering extensive areas ($\sim 10,000$ km$^2$, e.g., Sophocleous and Perkins, 2000; De Lange et al., 2014; Højberg et al., 2013), these models are calibrated only across specific portions of the system, thus challenging their predictive capabilities. Even in these cases, uncertainty associated with the estimated model parameters is usually overlooked. De Graaf et al. (2020) present a detailed geological reconstruction for the same domain analyzed by Maxwell et al. (2015). Due to computational constraints, their model could only simulate groundwater flow within selected portions of the domain and calibration of model parameters for the entire model domain was not achieved. Recently, Mather et al. (2022) introduced a three-dimensional data-driven model of continental-scale groundwater flow. It is noted that data-driven models are heavily constrained by the quantity and quality of available training data. In this context, groundwater flow and pressure data may not be as readily accessible as, for example, lithostratigraphic data (see e.g., Manzoni et al., 2023) across the entire domain, especially when considering large-scale scenarios. In general, a comprehensive calibration strategy encompassing the entire geographical extent of the model domain is still lacking.

Groundwater systems are inherently heterogeneous, thus rendering modeling of flow and transport processes in such complex domains prone to uncertainty. The latter stems from the (generally unknown) spatial distribution of medium properties, boundary conditions and/or forcing terms, and limited data availability. This issue could be addressed upon relying on a stochastic framework for model calibration (e.g., Neuman, 2003; Riva et al., 2009; Ye et al., 2010; Panzeri et al., 2015; Siena and Riva, 2020). However, stochastic model calibration presents significant challenges, particularly in terms of computational

cost when dealing with multiple source of uncertainty (e.g., Vrugt et al., 2008; Hendricks Franssen et al., 2009; Zhou et al., 2014). Although stochastic model calibration has become feasible at laboratory scales ($\sim 10^{-2} - 1$ m$^2$) and at experimental sites of limited areal extent ($\sim$1-100 km$^2$), the impact on the hydraulic response across large scale fields stemming from the inherent uncertainty plaguing our knowledge of the subsurface is still largely unexplored. In this framework, Bianchi Janetti et al. (2019, 2021) analyze how the uncertainty related to the characterization of the subsurface system affects the distribution of hydraulic heads and subsurface fluxes in a regional-scale hydrological setting ($\sim$ 1,000 km$^2$).

Here, we introduce and test a methodological approach for stochastic model calibration tailored to large-scale scenarios (exceeding 10,000 km²). Our proposed methodology combines a suite of tools that have not been previously employed to address groundwater modeling at such a vast scale and with such level of system complexity. These include (a) modeling the dynamics of groundwater flow across a three-dimensional setting, (b) embedding and analyzing in details interactions between rivers and aquifers, (c) relying on a probabilistic reconstruction of geological material distributions and attributes, (d) resting on multi-objective optimization techniques for stochastic calibration of large-scale groundwater models, and (e) performing a detailed informed global sensitivity analysis to assess the degree of spatial variability of the relative importance of uncertain model parameters therein. Through incorporation of these tools, our methodological and operational workflow yields a calibrated model that enhances our understanding of aquifer dynamics from a holistic perspective. It also reveals insights into the spatial pattern of the sensitivity of model outputs to model parameters. Results associated with the latter element can be employed to inform future data acquisition efforts to improve model parameterization and hydraulic head estimates. They also emphasize the need to balance model complexity with simplifications that might be required to tackle large-scale groundwater scenarios. The approach involves the development of a groundwater model that includes a probabilistic three-dimensional hydrogeological reconstruction of the investigated area. As stated above, we do so upon integrating the results illustrated by Manzoni et al. (2023). Specifically, we leverage on their probabilistic three-dimensional hydrogeological reconstruction, which enables us to infer the spatial distribution of geological properties at a scale that was previously unattainable. By incorporating this advanced hydrogeological reconstruction into our workflow, we address key challenges posed by uncertainties that are inherent to large-scale groundwater systems. Multi-objective optimization is a key step to assessing the way model parameters impact diverse system responses. This challenge is addressed upon relying on a coevolutionary optimization framework that is applied to a differential evolution optimization algorithm, thus ensuring effective control over the optimization process while preserving computational efficiency (e.g., Dagdia and Mirchev, 2020; Trunfio, 2015). The resulting algorithm is designed to handle multiple objectives and eliminates the need to assessing their relative weights within the overall objective function (Khan et al., 2022). The methodology we present is designed not only to increase our knowledge about the dynamics of large-scale systems but also to guide future data acquisition campaigns. The latter goal is attained by making use of an informed Global Sensitivity Analysis (GSA). We recall that GSA typically serves as a tool to assess the relative impact of uncertain model inputs on model outputs of interest (Morris, 1991; Campolongo et al., 2007; Razavi and Gupta, 2015; Pianosi et al., 2016; Dell'Oca et al., 2017). An informed GSA (Dell'Oca et al., 2020) is performed after (stochastic) model calibration and enables one to quantify the influence of residual (i.e., following model calibration) uncertainty associated with model

parameter estimates on predictions of system dynamics. This strategy aligns with our focus on tackling major challenges posed by large-scale subsurface flow scenarios. It offers critical insights on model functioning through quantification of the impact of model parameters on target model outputs. It also provides guidance on the identification of locations where acquiring additional information can enhance the accuracy of parameter estimates and ultimately constrain the uncertainty associated with model results.

The proposed methodological approach is then employed to analyze the largest groundwater system in Italy, corresponding to the Po River watershed. This region faces a variety of challenges, related to groundwater contamination (Guadagnini et al., 2020; Balestrini et al., 2021), sea water intrusion (Colombani et al., 2016), and water scarcity (Bozzola and Swanson, 2014). Thus, the design of comprehensive policies addressing risks to water quality across large scale groundwater systems of this kind is grounded on the implementation of a modeling framework capable of addressing the key patterns of groundwater flow at the scale of the entire domain (Giuliano, 1995; Nespoli et al., 2021).

The work is organized as follows. Sect. 2 provides an overview of the large-scale groundwater system we consider. The proposed methodology and workflow are illustrated in Sect. 3. Sect. 3.1 describes the modeling approach employed to assess groundwater recharge. Sect. 3.2 focuses on the large-scale groundwater model, which involves integrating data from multiple sources such as large-scale hydrogeological reconstruction, remote sensing, and global-scale databases. Sect. 3.3 delves into the inverse modeling approach employed and introduces a novel application of a coevolutionary algorithm to address the multi-objective function associated with large-scale hydrogeology settings. Sect. 3.4 describes the informed GSA approach. Key results are presented in Sect. 4, while Sect. 5 summarizes main findings and implications.

## 2 General Setting

Our analysis focuses on the Po River basin. Along with the Rhône and the Nile, the Po is one of the main Mediterranean rivers. With an average flow rate of about 1,500 $m^3$/s, it has more than 140 tributaries forming an intricate network of waterways that also intersects with a dense network of irrigation canals (ISPRA, 2010). This area (denoted as Po Plain, Pianura Padana) encompasses the largest and most exploited groundwater system across Italy, which provides fresh water to about 24 million residents (ISTAT, 2020). This area holds significant economic importance, contributing to nearly 40% of Italy's Gross Domestic Product. Due to the high density of industrial and agricultural activities, the system is facing a significant risk of overexploitation and possible exposure to multiple contaminants (AdB-Po, 2021). Seawater intrusion is also a potential negative issue in the coastal portion of the system (Kazakis et al., 2019; Antonellini et al., 2008).

As shown in Fig. 1, the domain is geographically bounded by the Adige River to the northeast and by the Adriatic Sea to the east, while its remaining boundaries encompass the mountain ranges of the Alps and the Apennines.

Our study is framed across the entire Po River District (AdB-Po, 2021). The latter covers approximately 87,000 $km^2$, spanning nine Italian Regions as well as the Swiss canton of Ticino and some valleys in the French and Swiss Alps (see Fig. 1). This area includes the entire catchment area of the Po River (~72,000 $km^2$). Main features of the study area vary from the high

peaks of the Alps and Apennines (with altitude exceeding 4,000 meters above sea level, steep slopes of more than 15%, and a population density of less than one inhabitant per km$^2$) to flat terrain and densely populated areas (with more than 2,000 inhabitants per km$^2$) (SEDAC, 2018). The district also experiences notable climatic differences. The lowland area is

130 characterized by a continental and temperate climate with moderate annual precipitation levels ranging from 600 to 900 mm (Morgan, 1973; Grimm et al., 2023). The Alps include a variety of climate zones at different elevations, corresponding to distinct biotic features. These zones are often characterized by multiple precipitation patterns, including both snow and rain (Elsasser and Bürki, 2002; Agrawala, 2007). The foothill (Prealpi) zone features highest cumulative precipitation levels, with an annual precipitation of 1500-2000 mm (Fratianni and Acquaotta, 2017; Morgan, 1973).

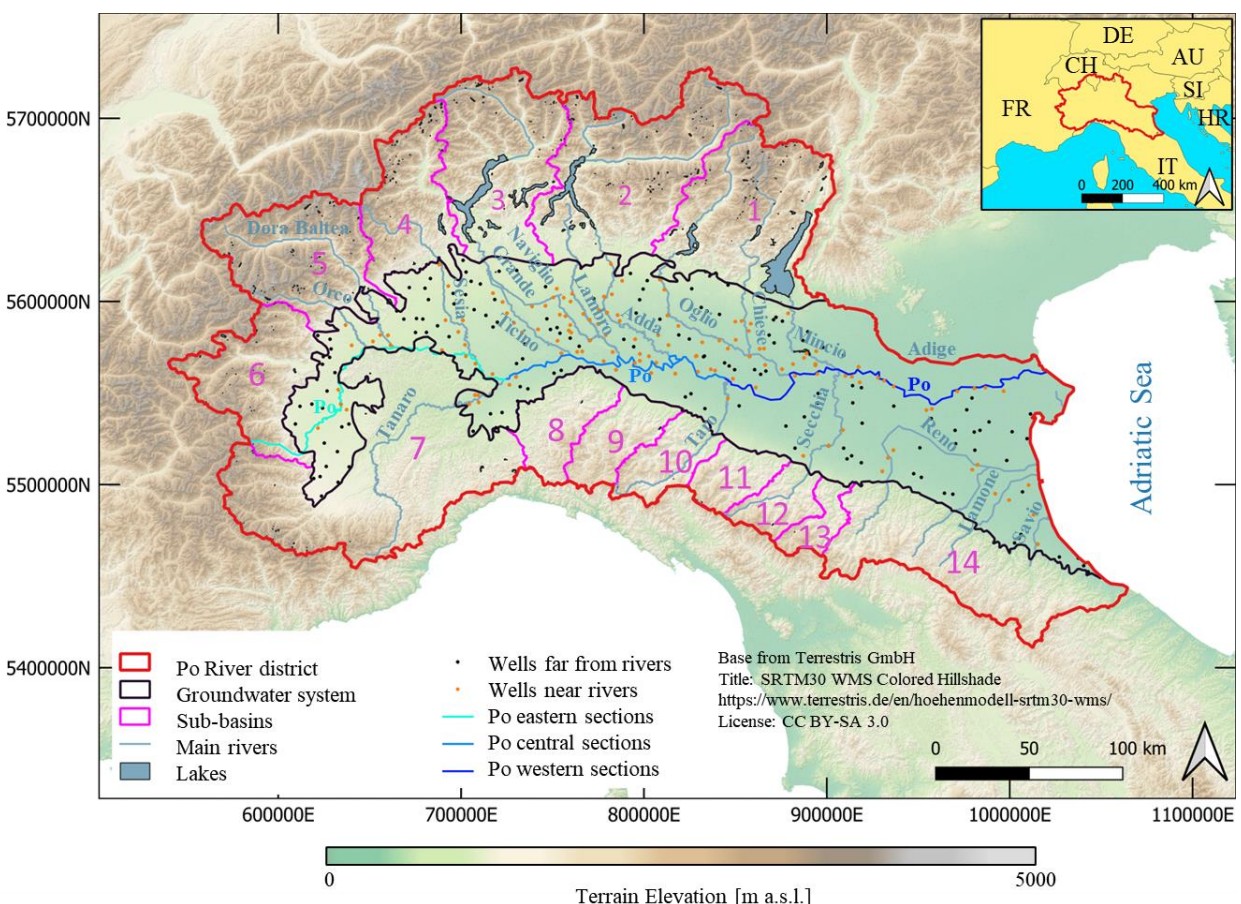

**Figure 1: Spatial location of the Po River District, including the Po Plain groundwater system and the sub-basins considered to assess lateral flow boundary conditions. Dots correspond to locations of wells for which head data are available. Coordinates Reference System (CRS) = ESRI:54012.**

## 3 Methods

Steady state groundwater flow across the large-scale system described in Sect. 2 is evaluated through a three-dimensional (3D) finite element model that we develop in the OpenGeoSys v. 6.4.1 (Bilke et al., 2022). We describe the methodology used for evaluating groundwater recharge in Sect. 3.1. This step is performed within the entire Po district (i.e., not only within the considered groundwater system) to assess (*i*) surface recharge within the domain and (*ii*) contributions of the surrounding basins to lateral flow exchanges with the domain. Sect.s 3.2, 3.3, and 3.4 include key details about the groundwater model, its calibration, and the informed GSA, respectively. Figure 2 illustrates the conceptual workflow of the proposed methodology.

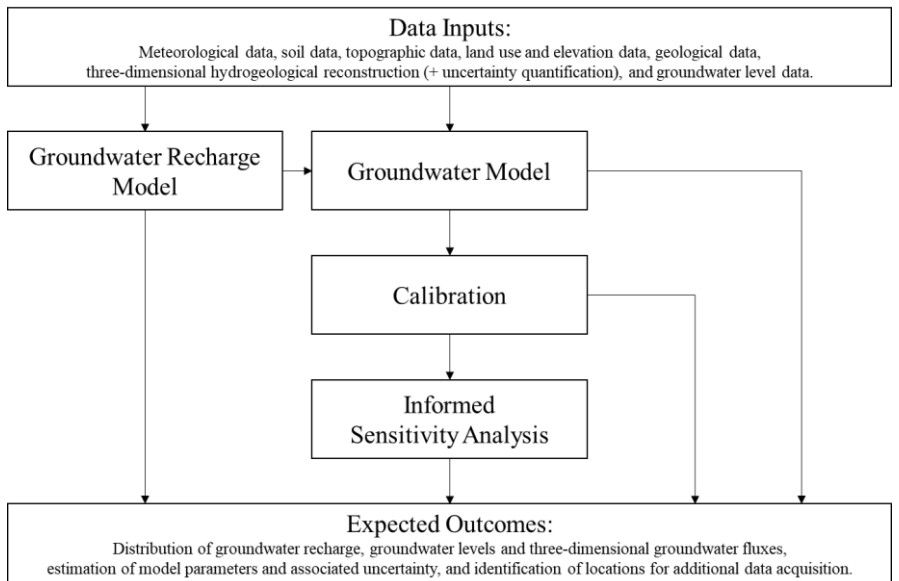

**Figure 2: Conceptual workflow for stochastic calibration and informed Global Sensitivity Analysis of large-scale groundwater models.**

### 3.1 Groundwater Recharge

We estimate the spatial and temporal variations of groundwater recharge at the scale of the entire Po district ($\sim$87,000 km$^2$). The study area is discretized into square cells with a spatial resolution of $250 \times 250$ m (resulting in approximately 1.4 million cells). Cell elevation data are obtained through the European Digital Elevation Model (ESA, 2019).

Groundwater recharge, $R$, is evaluated within each grid cell upon making use of the soil-water balance method of Thornthwaite (1948) and Thornthwaite and Mather (1955, 1957), as implemented in the widely used and tested (e.g., Zhang et al., 2016; Shuler et al., 2021; Roland et al., 2021) USGS SWB model Version 2.0 (Westenbroek et al., 2018), i.e.,

$$R = RG + SM + IRR + R_{in} - ET - R_{off} - (SWHC - SWC). \tag{1}$$

Here, $RG$ is the non-intercepted rain (rainfall reaching the ground); $SM$ is snowmelt; $IRR$ is irrigation; $ET$ is actual evapotranspiration; and $R_{in}$ and $R_{off}$ are overland inflow and outflow, respectively. The last term in Eq. (1) corresponds to the amount of water that can still be stored in the soil at a given time, $SWHC$ and $SWC$ being soil water holding capacity and soil water content, respectively. Equation (1) is solved with a temporal resolution equal to one day, covering the entire period from January (2010) to December (2019). Due to uncertainty of initial conditions, model results from January (2010) to December (2012) are discarded as they are associated with the warm-up periods of the hydrological model (see, e.g., Kim et al., 2018).

Meteorological data (such as precipitation and temperature) are obtained from the latest generation of ECMWF reanalysis data from ERA5 (Copernicus Climate Change Service (C3S), 2017). This dataset includes daily maximum and minimum temperatures evaluated at an elevation of two meters above ground, as well as precipitation data. The spatial distribution of the required soil information is collected from the global-scale maps of Poggio et al. (2021). To estimate the land cover type, we integrate crop type spatial distribution data from the EU CROP MAP (d'Andrimont et al., 2021) into the CORINE land cover map (European Environment Agency, 2018).

For the evaluation of $RG$, we account for a water interception budget. The latter represents the amount of precipitation that can be intercepted by vegetation. This interception budget varies across space depending on land use. Precipitation must exceed the intercepted amount in order to reach the soil and contribute to the soil water balance. The accumulation and melting term, $SM$, is evaluated on the basis of precipitation and maximum and minimum daily temperatures, as proposed by Dripps and Bradbury (2007). We recall that, according to previous studies (Farinotti et al., 2016), the investigated area receives a significant contribution from glacier melt. The irrigation term, $IRR$, is triggered only in the absence of precipitation during crop-specific irrigation period. It is evaluated (in each cell) by dividing the crop water need (i.e., the amount of water required to meet the evapotranspiration losses, considering the crop type and its growth stage) by the field application efficiency. We rely on the FAO 56 model (Allen et al., 1998) to assess the crop water need for 31 diverse types of crops identified in the area, for four different growth stages and their related periods. Due to lack of detailed space- and time-dependent irrigation data, here we use a constant field application efficiency value, set to its average national counterpart of 0.75 (Wriedt et al., 2009). For the evaluation of the actual evapotranspiration, potential evapotranspiration is first computed by (*i*) making use of the model provided by Hargreaves and Samani (1985) in non-irrigated regions and (*ii*) combining the Penman-Monteith model with the correction crop coefficient in cultivated areas (consistent with Allen et al., 1998). The latter method has been developed and widely applied for estimating evapotranspiration in irrigated soils. Actual evapotranspiration is then computed on the basis of the soil water content. If $SWC$ is larger than the potential evapotranspiration, $ET$ is equal to the potential evapotranspiration; otherwise, $ET = SWC$. Note that the Hargreaves-Samani and Penman-Monteith models are implemented without considering any corrections for wind effects, as the study area experiences weak surface winds, with an average wind speed of approximately 2 m/s (Bonafe' et al., 2012). Overland outflow (or surface runoff) is evaluated upon making use the Soil Conservation Service (SCS) Curve Number (CN) method (Mishra and Singh, 2003). Note that the estimation of $R_{off}$

requires the availability of maps of hydrological soil class and land cover type. Hydrological soil classes are assessed through the broadly used ROSETTA software (Schaap et al., 2001), that makes use of physical soil attributes such as clay and sand soil content as well as soil bulk density. Given the presence of very cold temperatures in different periods of the year for a
large portion of the study area, we include a runoff enhancement factor in the case of frozen ground, as proposed by Molnau and Bissell (1983). Finally, the overland inflow to a given cell is evaluated as the sum of $R_{off}$ values computed for the uphill neighbor cells in the previous time step iteration. A workflow of the recharge modeling approach is offered in Fig. 3. Detailed information regarding all input values here employed can be found in the open code repository (https://doi.org/10.5281/zenodo.10013442).

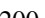

**Figure 3: Conceptual workflow of the Groundwater Recharge Model.**

## 3.2 Groundwater modeling approach

We build a large-scale groundwater model that covers an area of approximately 31,500 km² within the Po River district (see Fig. 1). The architecture of the subsurface system is assessed by curating information embedded in datasets from three distinct local authorities. In this sense, we obtain an original integration of data stemming from the hydrostratigraphic survey of Emilia-Romagna (Regione Emilia-Romagna, 1998), as well as from the regional water protection plans of the Lombardia (Regione Lombardia, 2016) and Piemonte (Regione Piemonte, 2022) Regions. These studies provide information on the lateral extent and the bottom surface of the depositional group that includes the groundwater system. This information has been obtained by local authorities upon integration of data from geological studies performed in the area. The evolution of the sedimentary basin, as controlled by geodinamic and climatological factors, is characterized by an overall regressive trend from Pliocene open marine facies to Quaternary marginal marine and alluvial deposits (Ricci Lucchi et al., 1982; Regione Emilia–Romagna and ENI, 1998; Regione Lombardia and ENI - Divisione AGIP, 2002). The aquifer system is characterized by a dense network of deep faults that influence the overall depth of the aquifers (Carcano and Piccin 2001), driving the variability of the groundwater system thickness from a few meters (close to the foothills) to more than 300 m (in the central and eastern portions of the plain). A continuous portion of virtually impermeable material can be found below the base surface. As already noted in Sect. 3.1, we employ the Digital Elevation Model (ESA, 2019) to determine the topographic map of land surface. This information enables us to determine the boundary of the groundwater system. Notably, the highest uncertainties in the hydrostratigraphic reconstruction model are found beyond the lateral boundaries of the groundwater system, where only a limited number of investigations is available.

We discretize the 3D subsurface domain through a hybrid mesh, as obtained within the Gmsh environment (Geuzaine and Remacle, 2009) through the OpenGeoSys Data Explorer GUI (Rink et al., 2013). The selected mesh enables us to capture the irregular shape of the boundaries of the investigated domain as well as its natural features while preserving the advantages of regular meshes for modeling layered geological systems. Domain discretization is performed according to a two-step approach. First, the ground surface is discretized using triangular elements with variable sizes (ensuring a maximum edge length of mesh elements of 5 km). Elements are adjusted to closely represent the ground surface as well as the irregular geometry of rivers and boundaries. The study employs a vertical discretization of the numerical grid that favors a balance between computational efficiency and the vertical distribution of geomaterials provided by the work of Manzoni et al. (2023). In this context, vertical discretization is finest near the surface, where thinner layers of geomaterials are documented. This is consistent with the availability of a high density of geological data at such depths, which has then facilitated identification of thin layers. Thus, the surface grid is then extruded along the vertical direction to the bottom surface underlying the whole groundwater system to create layers whose thickness increases with depth according to the following criteria: (*i*) layers maintain a constant thickness of less than 10 meters within the top 100 meters below the surface level; (*ii*) at depths comprised between 100 m and 200 m, the largest layer thickness is less than 20 meters, and (*iii*) a constant thickness of less than 40 m is maintained for layers corresponding to depths larger than 200 m.

To determine the types of geomaterials associated with each cell of the resulting grid, we rely on the detailed three-dimensional probabilistic hydrostratigraphic model developed for the Po River District by Manzoni et al. (2023). The dataset includes six macro categories (or geomaterials, denoted as gravel, sand, silt, clay, fractured rock, and rock) according to which the data associated with lithostratigraphic information across the area can be grouped. Manzoni et al. (2023) rely on a fine structured grid (resolution of $1000 \times 1000$ m along the horizontal plane and 1 m along the vertical direction) and evaluate the probability that each cell is associated with one of these six geomaterials. On these bases, we can evaluate the fraction of the $c$-th geomaterial that can be assigned to the $i$-th cell of our simulation grid, $f_{c,i}$, as

$$f_{c,i} = \frac{1}{N_i} \sum_j^{N_i} P_{c,j} \tag{2}$$

Here, $N_i$ denotes the number of cells associated with the hydrostratigraphic model of Manzoni et al. (2023) that are included in the $i$-th cell of our simulation grid and $P_{c,j}$ is the probability that the $c$-th category (or geomaterial) be assigned to cell $j$ of the above mentioned hydrostratigraphic model. Figure 4a depicts the percentage of simulation grid cells associated with given (color-coded) ranges of values for $f_{c,i}$ for each geomaterial category. One can note that clay and sand are the most abundant geomaterial categories identified across our domain. Otherwise, gravel is the most frequent among the remaining categories, being associated with about 15% of the simulation grid cells. Categories associated with silt or rock and fractured rock are found in a very limited proportion across the entire simulated domain.

Figure 4b illustrates the spatial distribution of the most probable geomaterial category within the Po River basin, as obtained by Manzoni et al. (2023). The reconstruction extends to a depth of 400 m below ground surface, covering the entire Po watershed and encompassing the full extent of the simulation grid.

We then assess the permeability of the $i$-th cell of the grid as

$$\bar{k}_i = \sum_c^{N_c} f_{c,i} k_c \qquad \text{with } N_c = 6 \tag{3}$$

where $k_c$ is the permeability of the $c$-th category; $k_c$ values are estimated through model calibration, while $f_{c,i}$ is provided as prior information (see Manzoni et al., 2023). Details regarding model calibration are illustrated in Sect. 3.3.

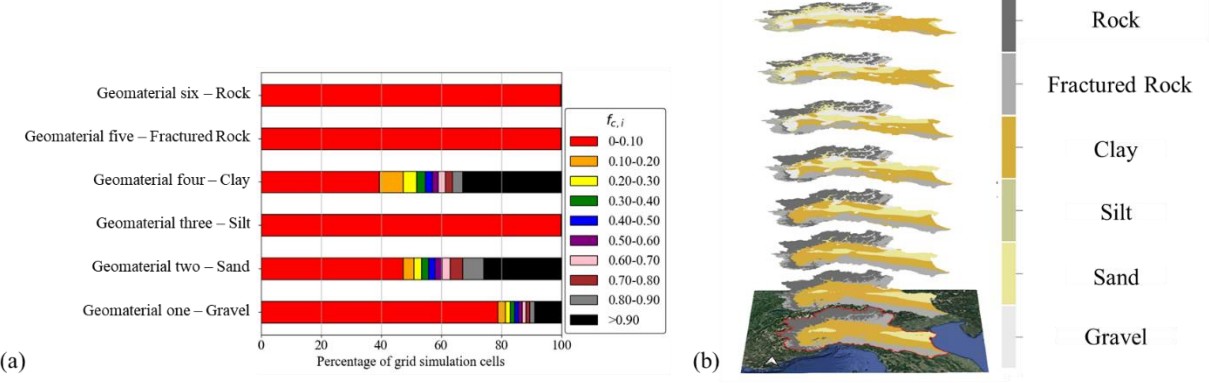

**Figure 4: (a) Percentage of grid cells characterized by given ranges of values of $f_{c,i}$ (Eq. 2); (b) Spatial distribution of modal categories obtained by Manzoni et al. (2023). Planar maps are selected at 5, 10, 25, 50, 100, 150, 200, and 350 m below ground surface.**

As boundary conditions, we set a constant hydraulic head, $h = 0$ m, along the coastline and a Cauchy boundary condition along the Adige River. Flow boundary conditions are imposed along the remaining lateral boundaries (see Fig. 1). Here, boundary fluxes are assigned using a mass balance analysis performed across the 14 main sub-basins surrounding investigate subsurface domain (denoted as sub-basins, $s = 1, 2, ..., 14$ in Fig. 1). The delineation of these sub-basins is provided by the Po River Basin Authority (AdB-Po, 2021). Inflow takes place through the vertical surface that extends from the ground surface to the aquifer base along the lateral extent of the aquifer system. Such lateral surface is typically characterized by a limited depth (only a few meters). Thus, lateral inflow is distributed uniformly across all layers of the lateral surface associated with each sub-basin. Making use of the results of Sect. 3.1, we evaluate within each of these sub-basins the average (in time) amount of water that infiltrates within a day as

$$Q_s = r_q R'_s S_s, \tag{4}$$

where $R'_s$ [L/T] is the (space-time averaged) recharge rate evaluated for the $s$-th sub-basin (with ground surface area of extent $S_s$) during the temporal window spanning the years 2013-2019. To account for possible exfiltration of infiltrated water or infiltration of water due to surface-groundwater interaction (e.g., river water infiltration), we also introduce a correction coefficient, $r_q$. The latter is set at a constant value for all sub-basins to avoid model overparameterization and is estimated through model calibration, as detailed in Sect. 3.3. Finally, we determine a uniform flow rate boundary condition for the $s$-th sub-basin as $q_s = Q_s/A_s$, $A_s$ corresponding to the lateral surface associated with a given sub-basin. We assign water flow rate boundary conditions at the ground surface of the domain upon considering the mean groundwater recharge (see Sect. 3.3) and domestic water use. To estimate the volumetric flow rate for domestic use, we rely on the public water supply data provided by the Italian National Institute of Statistics (ISTAT, 2020). This dataset contains values of flow rates (in $m^3$/year) employed for domestic purposes for each municipal administrative area as well as the share of domestic water associated with groundwater resources. Such data are available for the years 2012, 2015, 2018, and 2020. We then evaluate the average flow rate for each municipality on these bases. For a given municipality, domestic water fluxes associated with the use of groundwater resources is assessed upon evaluating the ratio of the total volumetric flow rate associated with groundwater extractions for drinking water to the surface area covered by the municipality itself (OpenStreetMap, 2021). Volumetric flow rates employed in the model are then obtained by multiplying the portion of the municipality area within the modeled domain by the domestic water flux. Due to the lack of comprehensive information regarding the location of extraction wells, we consider such a water flow rate as a distributed sink term located within the deepest layer of the simulation domain beneath each associated municipality. This assumption is grounded on the notion that drinking water wells are typically engineered to extract water from locations that are protected from potential contaminants that may infiltrate and pollute shallower regions of subsurface water bodies. Finally, we set Robin boundary conditions along the cells associated with the main 18 rivers comprised within the domain (see Fig. 1 for their location), i.e.,

$$Q_{r,i} = -C_{r,i}(h_i - h_{rs,i}),\qquad(5)$$

where $Q_{r,i}$ represent the water flow rate from the segment of the $r$-th river of length $L_{r,i}$ within the $i$-th grid cell to the groundwater systems; $C_{r,i}$ represents the riverbed conductance of segment $L_{r,i}$; and $h_i$ and $h_{rs,i}$ are the groundwater hydraulic head at cell $i$ and the elevation of the river stage of segment $L_{r,i}$, respectively. Each river is assigned a uniform specific conductance, $\alpha_r = C_{r,i}/L_{r,i}$, with the exception of the Po River. The latter is subdivided into three segments (see Fig. 1), each with a different specific conductance due to the varying geological characteristics, i.e.: (*i*) the eastern portion of the river flows over a geologic region mainly characterized by deltaic, floodplain, coastal, and wind deposits; (*ii*) the middle portion of the river flows over a geologic region mainly characterized by terraced alluvium and aeolian deposits; and (*iii*) the western portions of the river meander through hilly regions, which exhibit diverse geological features (Compagnoni et al., 2004). This subdivision leads to 20 distinct values of $\alpha_r$, estimated as detailed in Sect. 3.3.

## 3.3 Calibration Data and Inverse Modeling Strategy

In this section we report first the procedures applied to obtain calibration data from raw datasets (Sect. 3.3.1), and then we describe the main traits of the model inversion strategies (Sect. 3.3.2).

### 3.3.1 Data curation

Model parameters are estimated using time-averaged measurements of groundwater levels available across the domain and collected between January (2013) and December (2019). These data are available for the three main Italian Regions within which the groundwater system resides (i.e., Piemonte, Lombardia, and Emilia-Romagna). Available data are not homogeneous in terms of quantities monitored, temporal windows associated with data collection, and data format. Data curation is therefore a critical element to enable effective use of the available information. The resulting data set is here presented and employed for the first time. It serves as a basis upon which future studies aimed at further enhancing our knowledge of the hydrological functioning of this large-scale groundwater system and designing appropriate water management strategies therein can be developed.

Hydraulic head data have been collected with a sampling frequency of eight hours for the Piemonte Region (Agenzia Regionale per la Protezione Ambientale Piemonte, 2020). In the Emilia-Romagna and Lombardia Regions, the sampling frequency varies among wells, with an average approximately corresponding to 2 and 10 samples per year, respectively (Regione Emilia-Romagna, 2020; Regione Lombardia, 2021). We apply a filtering process to the raw data before combining the different datasets. To avoid seasonal biases, we exclude from the dataset wells that do not have at least one observation in two different seasons for each year within the given time range. Furthermore, we exclude observation wells affected by local operational activities. For the $N_{h_b} = 286$ remaining wells, whose locations are indicated in Fig. 1, we evaluate the average hydraulic head, $\bar{h}_l$ (with $l = 1, ..., N_{h_b}$), associated with the investigated period.

### 3.3.2 Model calibration

Model parameters are estimated through a multi-objective optimization approach. The latter is tied to the joint minimization of two objective functions formulated as

$$f_{N_{h_b}} = \sqrt{\frac{\sum_{l=1}^{N_{h_b}}(\overline{h_l}-h_l)^2}{N_{h_b}}}$$
(6)

and

$$f_{N_{h_r}} = \sqrt{\frac{\sum_{l=1}^{N_{h_r}}(\overline{h_l}-h_l)^2}{N_{h_r}}}$$
(7)

where $\overline{h_l}$ and $h_l$ denote observed and estimated hydraulic head at well $l$, respectively. Estimation of permeability of each geomaterial (i.e., $k_c$ in Eq. (3)) and of the correction coefficient (i.e., $r_q$ in Eq. (4)) entails minimizing Eq. (6) (considering all available hydraulic head data, $N_{h_b}$). To estimate the specific conductance of the riverbeds, $\alpha_r$ (with $r = 1, ..., 20$), we minimize Eq. (7) with $N_{h_r} < N_{h_b}$, where $N_{h_r}$ is the number of wells located within a maximum distance of 5 km from a river (see orange dots in Fig. 1). Including this constraint on the distance between a river and observation wells enables us to refine the estimation of $\alpha_r$ by considering only hydraulic head observations that are significantly impacted by the interconnection between the groundwater system and the rivers. Note that minimization of Eq. (6) and (7) is tantamount to relying on a Maximum Likelihood (ML) estimation approach assuming that measurement errors of hydraulic head are not correlated and can be described through a Gaussian distribution (Carrera and Neuman, 1986). The two objective functions to minimize are closely interconnected. We implement an enhanced variant of the Differential Evolution (DE) optimization method (Storn and Price, 1997) to effectively minimize both objective functions simultaneously. Here, we rest on a modified version of the Cooperative Coevolutionary Differential Evolution (CCDE) optimization algorithm proposed by Trunfio (2015). The implemented algorithm does not require defining a single weighted multi-objective function, as otherwise required by standard DE and standard CCDE. Thus, our approach eliminates the non-trivial task of determining the appropriate (relative) weights between each of the terms that constitute the multi-objective function (e.g., Dell'Oca et al., 2023). Resorting to a modified CCDE algorithm enables us to balance between simplicity and the efficiency documented for CAs when dealing with multi-objective fitness functions (Khan et al., 2022).

As nature-inspired optimization techniques, CAs draw upon principles of biological coevolution, where optimization problems are linked to coevolving species (Dagdia and Mirchev, 2020). CAs share similarities with Evolutionary algorithms, as their sampling mechanisms and dynamics are inspired by Darwin's Theory of Evolution. Just as species evolve based on their fitness to survive and reproduce within an environment, solutions within a search space evolve to achieve the minimum of an objective function (Simoncini and Zhang, 2019). Additionally, the coevolution principle considers that a change in one species can trigger changes in related species, thus leading to adaptive changes in each species (Khan et al., 2022). In this context, Eq.s

(6) and (7) represent optimization functions for two coevolving species. These are then optimized through the modified CCDE. Our algorithm differs from CCDE (Trunfio, 2015) primarily in the way we define the dimensions of the two species. Instead of employing random or dynamic grouping strategies (Zhenyu et al., 2008; Trunfio, 2015), we opt for a supervised grouping strategy linking one of the model parameters (i.e., riverbed conductance, $\alpha_r$) to one species and the remaining parameters to the other species.

We choose a modified version of Coevolutionary Differential Evolution (CCDE) over the widely used NSGA II (or its variant CC-NSGA-II) for our algorithm. Both these algorithms use a divide-and-conquer strategy and are effective for high dimensional optimization. However, while NSGA II relies on a genetic algorithm, our algorithm utilizes Differential Evolution (DE). According to Tusar and Filipic (2007), DE-based algorithms outperform GA-based algorithms in multi-objective optimization due to a more efficient exploration of the parameter space. This element is particularly critical when optimal solutions lie on parameter bounds or amidst many local optima.

Additionally, our implemented algorithm does not explicitly optimize a *front*, which is otherwise a central concept in NSGA-II. Instead, it focuses on optimizing individual objective function values without introducing a dominance concept considering both objectives. This approach leads to a single set of optimized parameters, thus simplifying the optimization process through a balance of the contribution of both objective functions.

The implemented algorithm is designed to address global optimization problems through alternate evolution of candidate solutions between the two different species. The algorithm uses mutation, crossover, and selection strategies to enhance the quality of solutions as detailed in the following. First, we introduce the populations of candidate solutions. For each of the two species (where *sp* takes the values of one or two, for species associated with Eq. (6) or (7), respectively), we consider a set of $N_S$ candidate solutions (or members), denoted as $\boldsymbol{S_{sp}} = \left[\boldsymbol{s}_{sp,1}, \ldots, \boldsymbol{s}_{sp,m}, \ldots, \boldsymbol{s}_{sp,N_s}\right]$. Following Storn and Price (1997), we set $N_S = 15 \times N_p$, $N_p$ being the number of parameters (i.e., $N_p = 7$ or 20 for Eq. 6 or Eq. 7, respectively). Initial candidate solutions are defined by randomly selecting parameter values from a parameter space whose extent is designed to encompass a broad range of possible solutions.

Members of the populations are combined and mutated to calculate the next generations of candidate solutions as follows. We start by computing a mutated vector for each $m$-th candidate solution of a species associated with the $k$-th iteration of the optimization algorithm (or generation) as:

$$\hat{\boldsymbol{s}}_{sp,m}^k = \boldsymbol{s}_{sp,m}^k + F\left(\boldsymbol{s}_{sp,a}^k - \boldsymbol{s}_{sp,b}^k\right), \tag{8}$$

Here, $F$ represents an algorithm parameter (termed differential weight) that is set equal to 0.5 and $\boldsymbol{s}_{sp,a}^k$ and $\boldsymbol{s}_{sp,b}^k$ (with $a \neq b \neq m$) correspond to two (randomly selected) members of the population. We then combine parameters of $\hat{\boldsymbol{s}}_{sp,m}^k$ and $\boldsymbol{s}_{sp,m}^k$ to determine the trial vector $\tilde{\boldsymbol{s}}_{sp,m}^k$: if a parameter of $\tilde{\boldsymbol{s}}_{sp,m}^k$ is selected for mutation, its value is taken from $\hat{\boldsymbol{s}}_{sp,m}^k$; otherwise, it is taken from $\boldsymbol{s}_{sp,m}^k$. We randomly choose the parameters of $\boldsymbol{s}_{sp,m}^k$ that will undergo mutation among the parameters associated with the *sp* species, with a probability of parameter mutation set to 0.5. We finally select the $m$-th candidate solution of the

$(k + 1)$-th generation, $\boldsymbol{s}_{sp,m}^{k+1}$, by comparing the trial member, $\tilde{\boldsymbol{s}}_{sp,m}^k$, and the $m$-th population member from the $k$-th generation, $\boldsymbol{s}_{sp,m}^k$, based on the following condition:

$$385 \quad \boldsymbol{s}_{sp,m}^{k+1} = \begin{cases} \tilde{\boldsymbol{s}}_{sp,m}^k, & if\, f_N\big(\tilde{\boldsymbol{s}}_{sp,m}^k\big) < f_N\big(\boldsymbol{s}_{sp,m}^k\big) \\ \boldsymbol{s}_{sp,m}^k, & if\, f_N\big(\tilde{\boldsymbol{s}}_{sp,m}^k\big) \geq f_N\big(\boldsymbol{s}_{sp,m}^k\big) \end{cases} \qquad \text{with } f_N = f_{N_{h_l}} \text{ or } f_{N_{h_r}}. \qquad (9)$$

The algorithm steps can be summarized as follows at a given iteration $k$:

1. Calculate a new generation $(k+1)$ of the first species using Eq.s (8)-(9) with $f_N = f_{N_{h_l}}$, while keeping the parameters of the second species fixed;

2. Transfer the parameter set with the best performance, $\boldsymbol{s}_{1,best}^{k+1}$, among the members of $\boldsymbol{s}_{1,m}^{k+1}$ to the second species;

3. Maintain the parameters of the first species as fixed while calculating $\boldsymbol{s}_{2,m}^{k+1}$ (the next generation of the second species), thus repeating step 1 for the second species with $sp = 2$ and Eq. (7);

4. Pass back to the first species the parameter set of the member in the second species with the best objective function value, $\boldsymbol{s}_{2,best}^{k+1}$;

5. repeat steps 1 to 4 until a stopping criterion is met.

The patience stopping criterion is here employed for both objective functions, i.e., the algorithm stops if no improvement in performance over 80 consecutive iterations (or epochs) is detected. Figure 5 illustrates the Pseudocode of the algorithm.

**Begin**

    Initialize $\boldsymbol{S_1} = \left[\boldsymbol{s_{1,1}}, \dots, \boldsymbol{s_{1,m}}, \dots, \boldsymbol{s_{1,N_{s1}}}\right]$ and $\boldsymbol{S_2} = \left[\boldsymbol{s_{2,1}}, \dots, \boldsymbol{s_{2,m}}, \dots, \boldsymbol{s_{2,N_{s2}}}\right]$.

    Evaluate the members of $\boldsymbol{S_1}$ using Eq. (6).

    Evaluate the members of $\boldsymbol{S_2}$ using Eq. (7).

    **While** stopping criteria are not met for both the species:

        **For** each $m$-th member of $\boldsymbol{S_1}$

            Create $\tilde{\boldsymbol{s}}_{1,m}^{k}$ by applying mutation and crossover.

            Evaluate $\tilde{\boldsymbol{s}}_{1,m}^{k}$ using Eq. (6).

            Select $\boldsymbol{s}_{1,m}^{k+1}$ according to Eq. (9).

            Select $\boldsymbol{s}_{1,best}^{k+1}$

        **EndFor**

        Fix the parameters associated with Eq. (6) equal to $\boldsymbol{s}_{1,best}^{k+1}$.

        **For** each $m$-th member of $\boldsymbol{S_2}$

            Create $\tilde{\boldsymbol{s}}_{2,m}^{k}$ by applying mutation and crossover.

            Evaluate $\tilde{\boldsymbol{s}}_{2,m}^{k}$ using Eq. (7).

            Select $\boldsymbol{s}_{2,m}^{k+1}$ according to Eq. (9).

            Select $\boldsymbol{s}_{2,best}^{k+1}$

        **EndFor**

        Fix the parameters associated with Eq. (7) equal to $\boldsymbol{s}_{2,best}^{k+1}$.

        k++

    **EndWhile**

    Return $\boldsymbol{s}_{2,best}^{k+1}$ and $\boldsymbol{s}_{1,best}^{k+1}$.

**End**

Figure 5: Pseudocode of the employed algorithm.

Finally, to quantify the residual (i.e., after calibration) uncertainty associated with each estimated model parameter, we compute the parameter estimation covariance matrix, $\boldsymbol{\Sigma}_N$, as

$$\boldsymbol{\Sigma}_N \Big/ \sigma_{h,N}^2 = [\mathbf{J}^\mathbf{T}\,\mathbf{J}]^{-1}, \qquad \text{with } N = N_{h_b}, N_{h_r} \tag{10}$$

where $\mathbf{J}$ is the Jacobian matrix (T denoting transpose) of size $[N \times N_p, N_p]$ and $\sigma_{h,N}^2$ is measurement error variance. The latter is generally unknown and can be computed a posteriori as detailed in Carrera and Neuman (1986). Matrix $\mathbf{J}$ contains the

derivatives of $h$ with respect to model parameters. These are evaluated at the end of the optimization procedure using a centered difference scheme.

### 3.4 Global Sensitivity Analysis

Global Sensitivity Analysis is performed in the surrounding of the parameter values obtained through model calibration. A GSA analysis provides valuable insights on the impact of parameter uncertainty on the simulated variable (i.e., hydraulic head values in our case). Furthermore, an informed GSA offers guidance about where new (hydraulic head) measurements can enhance the quality of parameter estimates. As a GSA metric, we rely on the Morris indices. These are defined through the introduction of elementary effects, $EE_{\theta_{p,n}}$,

$$EE_{\theta_{p,n}} = \frac{h\left(\theta_1,\dots,\theta_p+\Delta\theta_p,\dots,\theta_{N_p}\right)-h\left(\theta_1,\dots,\theta_p,\dots,\theta_{N_p}\right)}{\Delta\theta_p} \tag{11}$$

Here, $EE_{\theta_{p,n}}$ is the incremental ratio for the uncertain parameter $\theta_p$ computed along trajectory $n$ within the parameter space; and $\Delta\theta_p$ is an increment evaluated as proposed by Campolongo et al. (2007). The Morris index $\mu^*_{\theta_p}$ is then defined as

$$\mu^*_{\theta_p} = \frac{1}{M}\sum_n^M \left|EE_{\theta_{p,n}}\right|, \tag{12}$$

Here, $M$ represents the number of trajectories (i.e., the number of diverse parameter combinations) selected employing a radial-sampling strategy (Campolongo et al., 2007). Stable results have been obtained with $M = 500$, requiring $(M + 1)N_P$ forward model simulations. We recall that the absolute value in Eq. (12) prevents cancellation between positive and negative values of $EE_{\theta_{p,n}}$. Variations in the value of parameters associated with low values of $\mu^*_{\theta_p}$ induce negligible changes in $h$. Note that we evaluate $\mu^*_{\theta_p}$ at all spatial locations within the simulated domain. This enables us to create a three-dimensional spatial distribution of Morris indices, providing insights on the impact of each parameter on hydraulic head values across the entire domain.

### 4 Results and discussion

This Section is devoted to the discussion of the results related to groundwater recharge spatial distribution (Sect. 4.1), groundwater flow model calibration and simulations (Sect. 4.2), and global sensitivity analysis (Sect. 4.3).

We begin by examining the spatial distribution of groundwater recharge and its impacts on the groundwater flow model. Our discussion encompasses model parameterization results and the large-scale three-dimensional flow patterns obtained through the calibrated model. The insights gained from model calibration assist the definition of an informed parameter space for the subsequent GSA.

## 4.1 Groundwater Recharge

Figure 6 depicts (time-averaged, during the years 2013-2019) spatial distribution of estimated annual groundwater recharge. The highest rates of recharge are detected in the Northern part of the domain, which is characterized by high precipitation levels and permeable geomaterials (Poggio et al., 2021). The eastern area of the domain exhibits shallow groundwater conditions and low permeability geomaterials, resulting in reduced infiltration rates. These findings are consistent with the spatial distribution of groundwater recharge presented by Rossi et al. (2022). These authors estimate groundwater recharge in Italy using a water balance approach and open access data upon relying on a spatial resolution that is otherwise coarser that the one we consider (i.e., grid-cell resolution of 10×10 km). Their study places annual groundwater recharge for the Po River watershed at values ranging between 27 and 37 billion m$^3$ per year. Our calculated average annual groundwater recharge for the entire watershed for the period 2013-2019 corresponds approximately to 38 billion m$^3$ per year, thus being in line with the above-mentioned range.

Most of the groundwater recharge takes place in the mountain areas of the Po River district, only approximately 0.4 billion m$^3$ per year being received from the top surface of the aquifer. This result suggests that the main water inflow to groundwater is related to the lateral surface located close to the foothills.

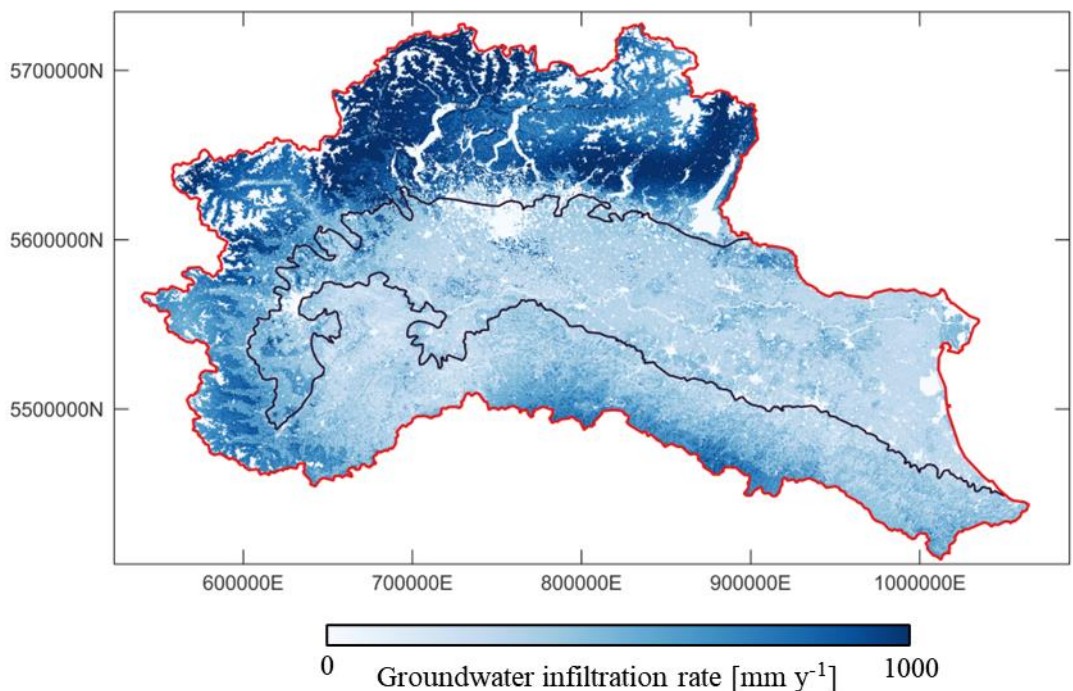

**Figure 6: Estimated mean annual groundwater recharge across the Po River District. Coordinates Reference System (CRS) = ESRI:54012.**

## 4.2 Groundwater Model

To ensure effective convergence of the CCDE algorithm, we rely on the set of metrics depicted in Fig. 7a-d. Note that the optimization algorithm leads to a converge of both objective functions ($f_{N_b}$ and $f_{N_r}$) in less than 50 iterations. The ensuing calibrated model is seen to display a remarkable degree of consistency with the system behavior observed across the domain (see Fig. 7c, d). The mean absolute error (in terms of hydraulic head) in the central and eastern areas of the Po plain is consistently low, averaging at about 4.5 m for these regions. Highest errors are observed near the foothill areas and in the planar areas of the Piemonte Region. Estimated model parameter values are listed in Table 1.

Results associated with the entries of the parameter estimation covariance matrices ($\mathbf{\Sigma}_{N_b}/\sigma^2_{h,N_b}$ and $\mathbf{\Sigma}_{N_r}/\sigma^2_{h,N_r}$) are depicted in Fig. 8a and Fig. 8b, respectively. As shown by the diagonal terms in Fig. 8a, the estimation variance of permeability ($k$) is higher for geomaterial categories five (fractured rock) and six (rock) as compared to the other ones. This result is related to the observation that these geomaterials are present in small amounts within the domain (see Fig. 4). Furthermore, there is a certain degree of negative correlation between permeability of geomaterial five and $r_q$. This finding is attributed to the fact that the simulation grid cells with the highest proportion of geomaterial five can be found in the mountainous areas and near the foothills (see Fig. 4b), which are close to the boundary where an inflow boundary condition is applied. Therefore, in these locations, an increase (or decrease) in the inflow across the boundaries can be obtained by increasing (or decreasing) both $k_5$ and $r_q$.

When considering riverbed conductance, it is observed that rivers with lower flows, such as the Chiese, Lamone, Savio, and Sesia (associated with parameters $\alpha_5$, $\alpha_{12}$, $\alpha_{13}$, and $\alpha_{17}$) rivers (see Fig. 1 for their planar location), exhibit the largest parameter estimation variance. In the central part of the Po River, the estimation variance of $\alpha_{10}$ is generally low. This suggests that the available data can effectively inform and provide valuable insights into the dynamics of river-groundwater interactions in this area. Conversely, estimates of parameters $\alpha_8$ and $\alpha_9$, characterizing the western and eastern portion of Po River, are associated with a high estimation variance. Additionally, a negative correlation can be observed between $\alpha_8$ and $\alpha_9$.

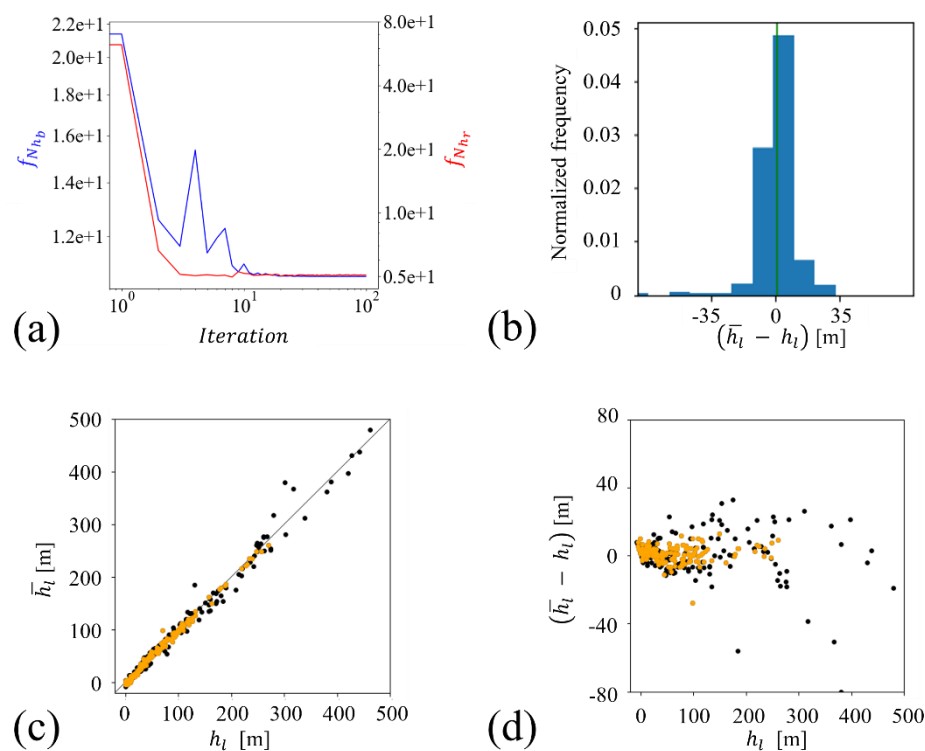

**Figure 7: (a) Convergence analysis of $f_{N_b}$ and $f_{N_r}$ (Eq.s (6) and (7), respectively); (b) normalized frequency distribution of differences between observed ($\bar{h}_l$) and simulated ($h_l$) hydraulic heads; (c) observed versus simulated hydraulic heads (values associated with the $N_{h_r}$ wells located close to the rivers are depicted in orange); (d) post-calibration residuals ($\bar{h}_l - h_l$) versus observed heads.**

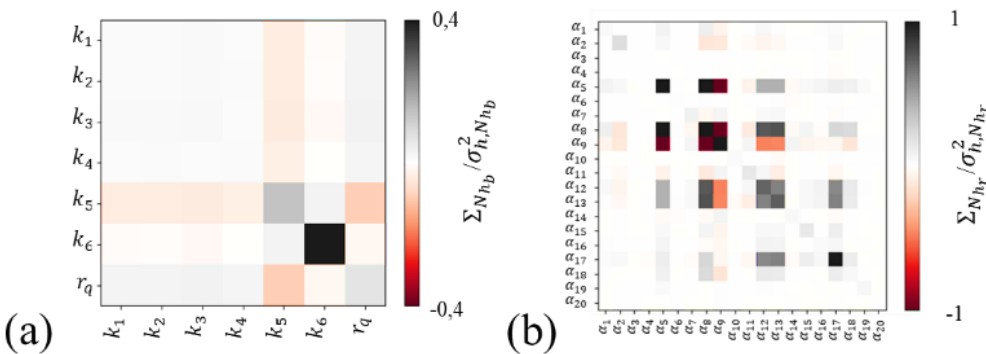

**Figure 8: Covariance matrix of parameter estimates related to (a) Eq. (6) and (b) Eq. (7).**

| Parameter | Description | Parameter estimate | Parameter range of variability |
|---|---|---|---|
| $k_1 \times 10^{-9}$ [m$^2$] | Permeability of geomaterial one | 1.02 | 0.64 – 1.61 |
| $k_2 \times 10^{-9}$ [m$^2$] | Permeability of geomaterial two | 3.83 | 2.24 – 6.08 |
| $k_3 \times 10^{-13}$ [m$^2$] | Permeability of geomaterial three | 2.24 | 1.41 – 2.55 |
| $k_4 \times 10^{-10}$ [m$^2$] | Permeability of geomaterial four | 1.57 | 0.99 – 2.49 |
| $k_5 \times 10^{-15}$ [m$^2$] | Permeability of geomaterial five | 2.12 | 1.34 – 3.37 |
| $k_6 \times 10^{-18}$ [m$^2$] | Permeability of geomaterial six | 5.04 | 3.18 – 8.00 |
| $r_q$ [-] | Lateral inflow correction coefficient | 0.99 | 0.79 – 1.19 |
| $\alpha_1 \times 10^{-5}$ [m$^2$s$^{-1}$] | Specific conductance of Adige | $8.18 \times 10^{-2}$ | $8.18 \times 10^{-4}$ – 8.18 |
| $\alpha_2 \times 10^{-5}$ [m$^2$s$^{-1}$] | Specific conductance of Secchia | $7.12 \times 10^{-2}$ | $7.12 \times 10^{-4}$ – 7.12 |
| $\alpha_3 \times 10^{-4}$ [m$^2$s$^{-1}$] | Specific conductance of Dora Baltea | $1.02 \times 10^{-2}$ | $1.02 \times 10^{-4}$ – 1.02 |
| $\alpha_4 \times 10^{-7}$ [m$^2$s$^{-1}$] | Specific conductance of Ticino | $1.10 \times 10^{-2}$ | $1.10 \times 10^{-4}$ – 1.10 |
| $\alpha_5 \times 10^{-3}$ [m$^2$s$^{-1}$] | Specific conductance of Chiese | $3.34 \times 10^{-2}$ | $3.34 \times 10^{-4}$ – 3.34 |
| $\alpha_6 \times 10^{-7}$ [m$^2$s$^{-1}$] | Specific conductance of Oglio | $2.44 \times 10^{-2}$ | $2.44 \times 10^{-4}$ – 2.44 |
| $\alpha_7 \times 10^{-5}$ [m$^2$s$^{-1}$] | Specific conductance of Tanaro | $3.04 \times 10^{-2}$ | $3.04 \times 10^{-4}$ – 3.04 |
| $\alpha_8 \times 10^{-2}$ [m$^2$s$^{-1}$] | Specific conductance of Po - western section | $2.95 \times 10^{-2}$ | $2.95 \times 10^{-4}$ – 2.95 |
| $\alpha_9 \times 10^{-2}$ [m$^2$s$^{-1}$] | Specific conductance of Po - eastern section | $1.34 \times 10^{-2}$ | $1.34 \times 10^{-4}$ – 1.34 |
| $\alpha_{10} \times 10^{-4}$ [m$^2$s$^{-1}$] | Specific conductance of Po - central section | $1.16 \times 10^{-2}$ | $1.16 \times 10^{-4}$ – 1.16 |
| $\alpha_{11} \times 10^{-5}$ [m$^2$s$^{-1}$] | Specific conductance of Reno | $2.90 \times 10^{-2}$ | $2.90 \times 10^{-4}$ – 2.90 |
| $\alpha_{12} \times 10^{-5}$ [m$^2$s$^{-1}$] | Specific conductance of Lamone | $5.53 \times 10^{-2}$ | $5.53 \times 10^{-4}$ – 5.53 |
| $\alpha_{13} \times 10^{-2}$ [m$^2$s$^{-1}$] | Specific conductance of Savio | $7.50 \times 10^{-2}$ | $7.50 \times 10^{-4}$ – 7.5 |
| $\alpha_{14} \times 10^{-5}$ [m$^2$s$^{-1}$] | Specific conductance of Adda | $3.20 \times 10^{-2}$ | $3.20 \times 10^{-4}$ – 3.20 |
| $\alpha_{15} \times 10^{-5}$ [m$^2$s$^{-1}$] | Specific conductance of Taro | $1.83 \times 10^{-2}$ | $1.83 \times 10^{-4}$ – 1.83 |
| $\alpha_{16} \times 10^{-8}$ [m$^2$s$^{-1}$] | Specific conductance of Mincio | $7.3 \times 10^{-2}$ | $7.30 \times 10^{-4}$ – 7.30 |
| $\alpha_{17} \times 10^{-2}$ [m$^2$s$^{-1}$] | Specific conductance of Sesia | $2.16 \times 10^{-2}$ | $2.16 \times 10^{-4}$ – 2.16 |
| $\alpha_{18} \times 10^{-4}$ [m$^2$s$^{-1}$] | Specific conductance of Orco | $1.79 \times 10^{-2}$ | $1.79 \times 10^{-4}$ – 1.79 |
| $\alpha_{19} \times 10^{-4}$ [m$^2$s$^{-1}$] | Specific conductance of Lambro | $4.39 \times 10^{-2}$ | $4.39 \times 10^{-4}$ – 4.39 |
| $\alpha_{20} \times 10^{-6}$ [m$^2$s$^{-1}$] | Specific conductance of Naviglio Grande | $1.03 \times 10^{-2}$ | $1.03 \times 10^{-4}$ – 1.03 |

**Table 1: Uncertain model parameters, associated estimated values resulting from model calibration and intervals of variability employed in the GSA.**

Figure 9 offers an overview of the three-dimensional distribution of permeability values across the subsurface domain. Figure 9a depicts the frequency distribution of the estimated permeabilities. These results reveal three dominant modes (or peaks) in the distribution. These are characterized by a frequency that is one order of magnitude higher with respect to the rest of

permeability values. This element suggests that the subsurface domain can be conceptualized (at this large scale) as a block-heterogeneous system comprising three main macro-areas, each of these being characterized by a mildly heterogeneous spatial distributions of permeability values. In this sense, the extent of each of these areas is assessed on the basis of the distribution of geomaterials, which in turn drives the spatial distribution of permeability. The spatial arrangement of these macro areas is consistent with the distribution of the three main sediment types indicated in the Italian Geological Map (Compagnoni et al., 2004) within the Po Plain (see Fig. 9c). Figure 9a provides an appraisal of the spatial distribution of the three macro-areas by means of envelopes obtained through projection of their otherwise three-dimensional shape on a two-dimensional plane. This visualization is complemented by Fig. 9b, which depicts a qualitative representation of the vertical distribution of $\log(k)$ along selected cross-sections (vertical exaggeration of 200). Access to a detailed grid of the three-dimensional distribution of permeability values is available through the code and data repository (https://doi.org/10.5281/zenodo.10013442).

The first macro-area, associated with the lowest permeability values within the modeled domain, generally corresponds to the south-eastern portion of the alluvial plain (Adriatic sector). Here, finer and less permeable sediments constitute the main features associated with geological deposition processes. The second macro-area is primarily located near the northern and western boundary, adjacent to the Alpine foothill areas, and is characterized by intermediate permeability values. Additional smaller areas with conditions similar to the Alpine foothills can be identified in the foothill areas of the Apennines. Note that, according to Éupolis Lombardia (2016), the planar area adjacent to the foothills in the Lombardia Region is very heterogeneous and features a series of highly permeable layers interspersed with less permeable layers. This is consistent with the intermediate range of permeability values obtained within our large-scale domain through model calibration. The third macro-area is characterized by high permeability values. It spans the entire depth of the system in the central-southern portion of the plain while it does not reach the surface in the northeastern part of the domain. This area is influenced by the deposits formed by the presence of the Po River.

As shown in Fig. 10a, hydraulic heads exhibit a higher gradient on the western side of the domain. This behavior can be attributed to the shallow depth of the aquifer and to the steep gradient of the domain bottom in this area. Figure 10b illustrates the way velocity magnitude and pattern are influenced by the three-dimensional distribution of the geomaterials and the thickness of the domain. As exemplified in section A-A', our results document that subsurface flow can be considered as chiefly two-dimensional (i.e., vertical flow is negligible) across regions where the groundwater system is very thin, and the bottom is fairly parallel to the ground surface. This is especially evident in the steepest areas within the domain. Otherwise, velocity distributions across sections B-B' and C-C' exhibit marked three-dimensional characteristics in terms of flow. With reference to section C-C', we note that lower permeability close to the domain bottom results in reduced groundwater fluxes, as compared to the other sections. Additionally, the bottom right side of section B-B' documents the impact of low-permeability lenses on the local three-dimensional patterns of fluxes (in terms of magnitude and direction). Finally, Fig. 10b documents spatial variability of permeability and groundwater flow across three selected vertical cross-sections near the rivers, highlighting effects of river-groundwater interactions.

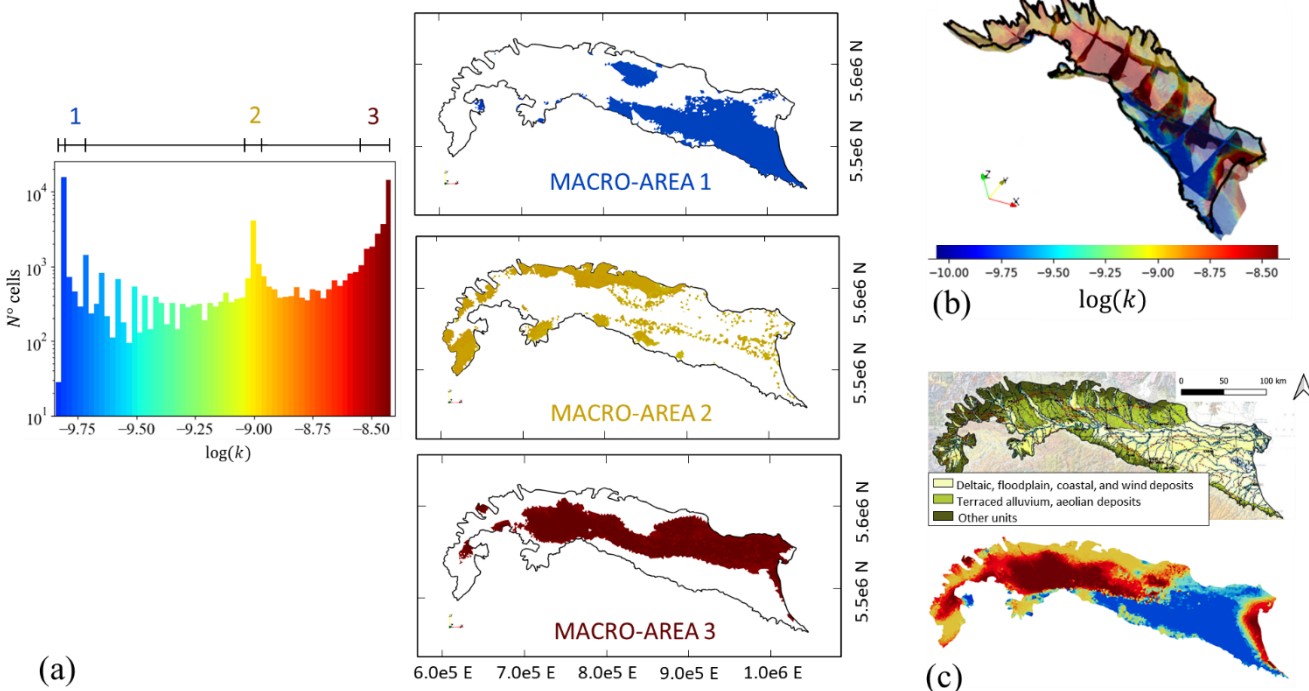

**Figure 9: (a)** Frequency distribution of natural logarithm of permeability, $\log(k)$ ($k$ expressed in m$^2$), estimates and spatial distribution of the three macro-areas corresponding to envelopes obtained through projection of their otherwise three-dimensional shape on a two-dimensional plane; **(b)** vertical distribution of $\log(k)$ along selected cross-sections (vertical exaggeration = 100); and **(c)** visual comparison between the spatial distribution of permeability estimates across the model top layer and the distribution of the three main sediment types indicated in the Italian Geological Map (Compagnoni et al., 2004) within the Po Plain.

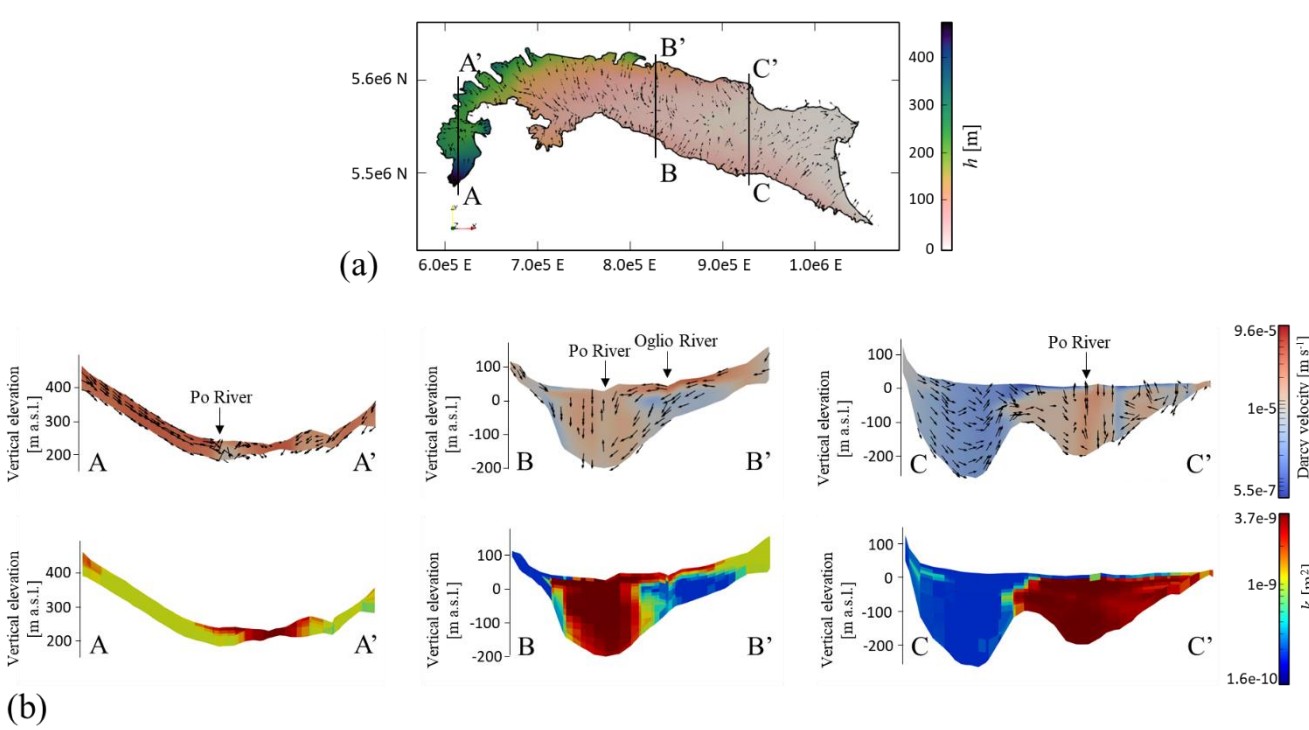

**Figure 10: Main groundwater flow model outputs: (a) hydraulic head and overall direction of groundwater fluxes across the top layer of the model; (b) magnitude and overall direction of groundwater fluxes and permeability distribution along cross-sections A-A', B-B', and C-C'(vertical exaggeration = 200).**

### 4.3 Global Sensitivity Analysis

Ranges of parameter variability employed for the GSA are listed in Table 1. These are selected to allow for (approximately) a 100% variability in permeabilities values, while values of parameters $\alpha_r$ ($r = 1, ..., 20$) can vary by four orders of magnitude. This choice enables us to account for the extensive uncertainty associated with the quantification of the interconnections between subsurface and surface water bodies, as these variables are typically not monitored in the field.

Figure 11 depicts values of $\mu^*_{\theta_p}$ associated with geomaterial permeability and correction coefficient $r_q$. These results suggest

that permeability values of geomaterial categories three, five, and six have a negligible impact on the spatial distribution of hydraulic heads. We recall that categories three, five and six are detected only in a limited amount within the modeled domain (see Fig. 4). As expected, permeability of geomaterial one (gravel) significantly influences simulation results in the foothills of the western portion of the domain, while of category four (clay) primarily affects simulation results in the southeastern portion of the domain. These results are in line with the spatial distribution associated with two lithologies. Category two

(sand) displays a noticeable impact on the hydraulic head distribution across the entire domain, which aligns with the observation that it is a widely available geomaterial within the system spanning from west to east. Finally, parameter $r_q$

significantly impacts hydraulic heads within all foothill areas, where lateral flow enters the groundwater system. As expected, its importance gradually decreases moving away from the boundary. The influence of permeability and $r_q$ significantly diminishes near the main rivers, where the flow field is primarily affected by parameters related to riverbed conductance (Fig.

12). Most of the riverbed conductance values can only affect hydraulic head estimates close to the rivers. This enables us to quantify the extent of the river influence on groundwater flow and further supports our calibration strategy, i.e., the use of the designed multi-objective optimization approach.

Rivers with the highest flow rates, such as the Adige ($\alpha_1$), Ticino ($\alpha_4$), Oglio ($\alpha_6$), Reno ($\alpha_{11}$), Adda ($\alpha_{14}$), and the central section of the Po River ($\alpha_{10}$), exhibit the highest values of $\mu^*_{\theta_p}$. Rivers like the Chiese River ($\alpha_5$) and the western ($\alpha_8$) and

eastern part of the Po River exhibit limited impact on simulated hydraulic head fields, partially due to their proximity to specific boundaries. These boundaries primarily influence groundwater flow through lateral boundary conditions (see Sect. 4.2), thus shadowing the effect of river-groundwater exchanges.

In Italy, irrigation channels have been documented to operate with efficiencies ranging from 0.43 to 0.6 (Wriedt et al., 2009). Then, a significant amount of the water lost from these channels enters the groundwater system. In this context, channels and

rivers such as the Naviglio Grande and the Lambro (associated with $\alpha_{19}$ and $\alpha_{20}$, respectively) show a significant influence on the local hydraulic head distribution, even as they are characterized by a generally low flow rate. This is related to the observation that they are located in an area with a dense irrigation channel network (De Caro et al., 2020) and their contribution to groundwater flow includes the cumulative effect of a high number of small irrigation channels. Fig. S1 (see supplementary material) illustrates the portions of the rivers recharging or draining the aquifer.

It is worth noting that all Morris indices display only modest variability along the vertical direction. The complete three-dimensional spatial distribution of $\mu^*_{\theta_p}$ and a grid containing 10 cross-sections highlighting our findings about the vertical variability of Morris indexes can be accessed in an open-source Visualization Toolkit (VTK) format for structured grids (Schroeder et al., 2006). These data are available in the code and data repository (https://doi.org/10.5281/zenodo.10664413).

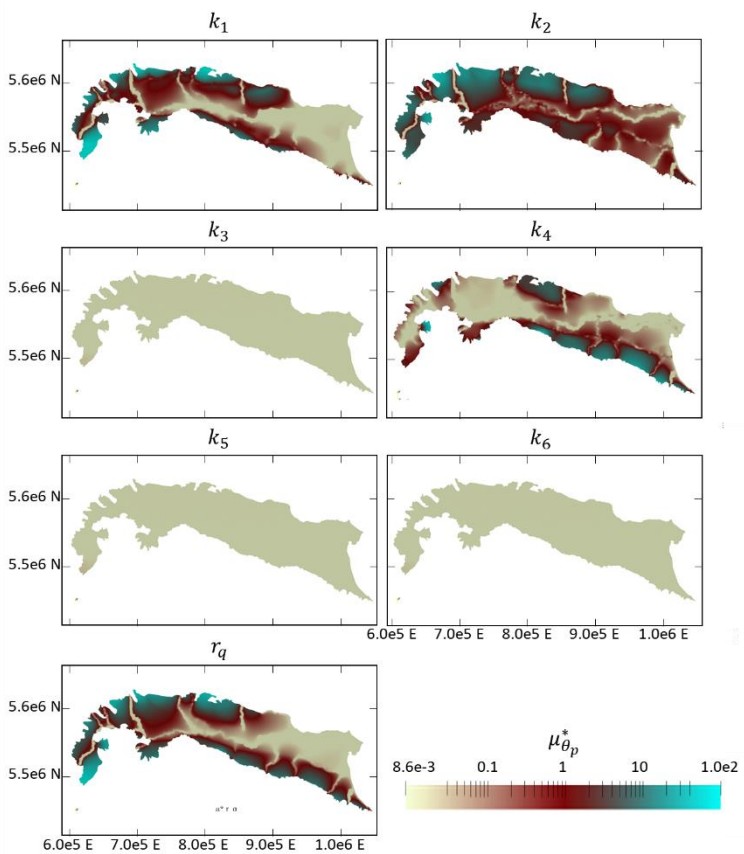

Figure 11: Spatial distribution of Morris indices related to geomaterial permeabilities ($k_1$, …, $k_6$) and correction coefficient $r_q$ across the top layer of the model.

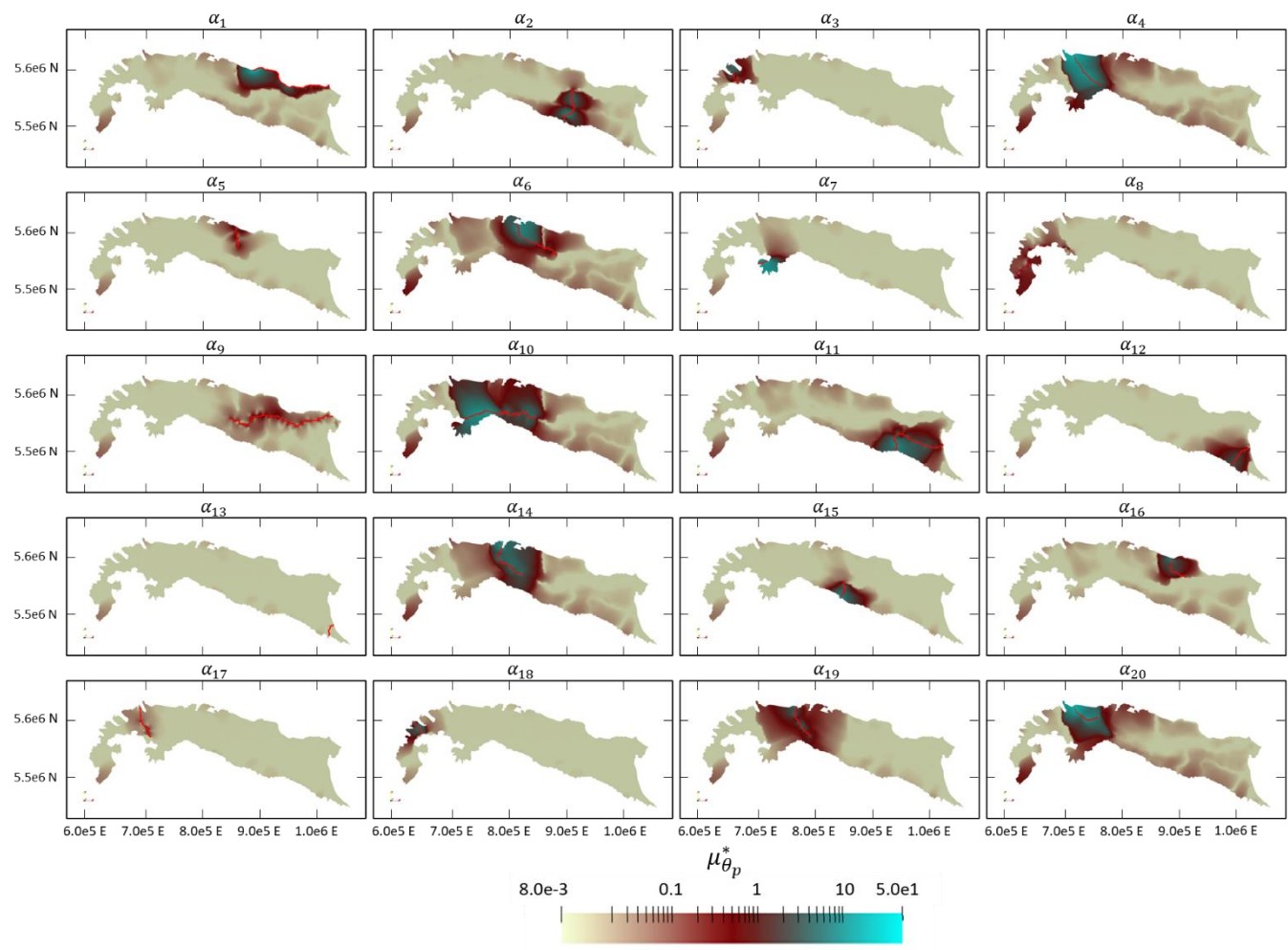

**Figure 12: Spatial distribution (across the top layer of the model) of Morris indices related to specific conductance of the riverbeds.**

## 5 Conclusions

The study introduces a comprehensive methodology that combines advanced numerical and data analysis methods, such as multi-objective optimization, informed GSA, and three-dimensional groundwater modeling, to analyze subsurface flow dynamics across large-scale domains. We support the suitability of the proposed approach to assess large-scale complex groundwater systems by employing it to analyze the main features of Italy's largest groundwater system, which is set within the Po River watershed. Our work leads to the following major conclusions.

1. Groundwater recharge is evaluated across the analyzed large-scale system upon relying jointly on remote sensing information and on-site data on land use main soil properties and attributes. While our results are overall consistent with prior findings across the area based on a global water balance approach (Rossi et al., 2022), they are otherwise associated with an enhanced spatial resolution. As such, they provide the basis for future applications aimed at delineating areas associated with vulnerability of the groundwater resource.

2. A coevolutionary algorithm is successfully employed for the calibration of our large-scale groundwater system model. Our approach allows differentiating the use of data depending on the spatial location of the observation wells. Notably, our approach is tailored to separate calibration of riverbed conductance, thus addressing surface-ground-water interactions with a dedicated optimization. Casting model calibration within a stochastic context yields quantification of the residual (i.e., after calibration on available information) uncertainty associated with model parameters. This ultimately enables one to identify model parameters whose estimates are associated with large uncertainty (as rendered through estimation variance) on the basis of the available dataset. In our scenario, the resulting model parameterization enables us to subdivide the domain into three macro-areas, each characterized by mild spatial heterogeneity of permeability. The spatial arrangement of these areas is in line with the distribution of sediment types documented by available geological maps associated with the studied domain (Compagnoni et al., 2004). While relying on a characterization of the system through a block heterogeneous conceptual picture is consistent with the scale tackled in our study, a detailed assessment of conductivity heterogeneous patterns might be required when targeting local scale settings. The latter can then be nested in the context of the large-scale patterns documented in our study.

The calibrated model enables us to identify three-dimensional flow patterns, as driven by the (three-dimensional) heterogeneous distribution of geomaterials across the subsurface. This represents a significant advancement as compared to commonly developed large-scale models based on two-dimensional geological maps.

3. Global Sensitivity Analysis (GSA) quantifies the relative importance of uncertain model parameters on a target model output (i.e., hydraulic heads) across the whole domain. Our results document the spatially heterogeneous distribution of global sensitivity metrics associated with model parameters, thus providing information about where the acquisition of future information could contribute to enhance the quality of groundwater flow model parameterization and constrain hydraulic head estimates. Our findings suggest that the features of the foothills (an area that is highly unexplored to date, as compared to lowland areas) should be subject to additional investigation to improve the quality of hydraulic head estimates. Furthermore, GSA results allow identifying rivers where information on water exchange with groundwater could be beneficial to improve piezometric characterization.

**Code and data availability**

All data and codes are available on the following repositories: https://doi.org/10.5281/zenodo.10664413 and https://doi.org/10.5281/zenodo.10013442.

**Author contributions**

AM, GMP, LG, AG, MR Conceptualization; AM Data curation; AM, GMP, LG, AG, MR Formal analysis AG, MR Funding acquisition; AM, GMP, LG, AG, MR Methodology; GMP, AG, MR Project administration; AM Software; GMP, LG, AG, MR Supervision; AM, GMP, LG, AG, MR Validation; AM, GMP, LG, AG, MR Visualization; AM, GMP, LG, AG, MR Writing - original draft preparation; AM, GMP, LG, AG, MR Writing - review & editing.

**Competing interests**

Some authors are members of the editorial board of Hydrology and Earth System Sciences journal.

**Financial support**

Funding. M. Riva acknowledges funding from the National Recovery and Resilience Plan (NRRP), Mission 4 Component 2 Investment 1.4 - Call for tender No. 3138 of December 16, 2021, rectified by Decree n.3175 of December 18, 2021 of Italian Ministry of University and Research funded by the European Union - NextGenerationEU; Project code CN_00000033,
Concession Decree No. 1034 of June 17, 2022 adopted by the Italian Ministry of University and Research, CUP D43C22001250001, Project title "National Biodiversity Future Center - NBFC". Support from Water Alliance (Acque di Lombardia) is also acknowledged. A. Manzoni and A. Guadagnini acknowledge funding from the European Union's Horizon 2020 research and innovation programme under the Marie Skłodowska-Curie grant agreement No 872607 in the context of the coordinated research program RECYCLE. G.M. Porta acknowledges funding from NRPP Next Generation EU Investment
1.1 through the PRIN project "Uncertainty Quantification of coupled models for water flow and contaminant transport"

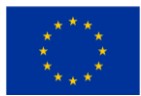

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
