# Peer review of "A Comprehensive Framework for Stochastic Calibration and Sensitivity Analysis of Large-Scale Groundwater Models"

_Hydrology and Earth System Sciences, 2023_

## Author Comment (AC1)

POLITECNICO DI MILANO

**DEPARTMENT OF CIVIL AND ENVIRONMENTAL ENGINEERING**

Piazza Leonardo da Vinci, 32 I-20133, MILANO (Italy)

27th February 2024

**Re: Response to Comments of Reviewer #1 -** *A Comprehensive Framework for Stochastic Calibration and Sensitivity Analysis of Large-Scale Groundwater Models* Submission to *Hydrology and Earth System Sciences*

We appreciate the efforts the Reviewer has invested in our manuscript. Following is an itemized list of the comments together with our response to each. Comments are listed in black italic font and our responses in blue font. Proposed revisions to the original text are in red fonts.

Sincerely,

Andrea Manzoni, Giovanni Michele Porta, Laura Guadagnini, Alberto Guadagnini, Monica Riva

general comments:

*The topic of the manuscript is of high relevance for the management of large aquifer systems. The presented approach provides a very suitable methodology to develop a groundwater model with predictive potential for a complex heterogeneous large-scale groundwater system. Such a model supports the understanding of the system dynamics and enables to identify parameters impacting diverse system responses. For the first time stochastic calibration and informed global sensitivity analysis are used to calibrate the groundwater model of the main aquifer system in the Po River watershed. The potential of the presented methodology is well concluded.*

We thank the Reviewer for the thorough evaluation and positive feedbacks on our manuscript. We appreciate the recognition of the relevance of our work and modeling approach in the context of management of large aquifer systems. We will carefully address the suggested revisions to further improve the quality of the manuscript.

*The manuscript is well structured and well readable. The title clearly reflects the contents of the paper. The abstract provides a concise and complete summary. The scientific methods and assumptions are valid and clearly outlined.*

*The authors give proper credit to related work and clearly indicate their own contribution. The number and quality of references appropriate.*

*I recommend the publication only after the revisions described below.*

specific comments:

In the following I would like to recommend several revisions in order to improve the manuscript.

**Comment #1:**

*The lateral extent and the base of the groundwater system should be described more clearly (l. 180/181). This should cover a more detailed the description of the interfaces with the sub-basins (l. 212/213) particularly the vertical distribution inflow boundary condition (l. 219). Furthermore, the basic geologic concept behind the vertical discretization is missing.*

**Answer #1:**

We thank the Reviewer for the valuable comment. We have taken steps to address the suggestions to enhance the quality and clarity of the large-scale groundwater model setup.

Details about the geometry of the groundwater system are accessible through the geological databases maintained by regional environmental authorities and listed in the original manuscript. We rely on location of boundaries and bottom of the modeled geometry that have been determined by local authorities, who integrated information from geological studies performed in the area. To enrich the information pertaining to the geometry, in the revised manuscript we will include additional details from some studies where the estimated location of the boundaries and of the base of the aquifer system within the study area are discussed. We will also expand on the description of the interfaces with the sub-basins focusing on the vertical distribution of the inflow boundary condition. We will also revise the manuscript to include a detailed explanation of the main concepts underpinning the vertical discretization employed in the numerical flow model.

We will incorporate the following modifications and additions to the revised manuscript in Section 3.2.

"The architecture of the subsurface system is assessed by curating information embedded in datasets from three distinct local authorities. In this sense, we obtain an original integration of data stemming from the hydrostratigraphic survey of Emilia-Romagna (Regione Emilia-Romagna, 1998), as well as from the regional water protection plans of the Lombardia (Regione Lombardia, 2016) and Piemonte (Regione Piemonte, 2022) Regions. These studies provide information on the lateral extent and the bottom surface of the depositional group that includes the groundwater system. This information has been obtained by local authorities upon integration of information from geological studies performed in the area. The evolution of the sedimentary basin, as controlled by geodinamic and climatological factors, is characterized by an overall regressive trend from Pliocene open marine facies to Quaternary marginal marine and alluvial deposits (Ricci Lucchi et al., 1982; Regione Emilia–Romagna and ENI, 1998; Regione Lombardia and ENI - Divisione AGIP, 2002). The aquifer system is characterized by a dense network of deep faults that influence the overall depth of the aquifers (Carcano and Piccin 2001), driving the variability of the groundwater system thickness from a few meters (close to the foothills) to more than 300 m (in the central and eastern portions of the plain). A continuous portion of essentially impermeable material can be found below the base surface.

…

Inflow takes place through the vertical surface that extends from the ground surface to the aquifer base along the lateral extent of the aquifer system. Since such lateral surface is typically characterized by a limited depth (only a few meters), lateral inflow is distributed uniformly across portions of lateral surface associated with each sub-domain.

...

This study employs a vertical discretization of the numerical grid that favors a balance between computational efficiency and the vertical distribution of geomaterials provided by the study of Manzoni et al. (2023). In this context, the vertical discretization is then finest closer to the surface, where thinner geomaterial layers are documented, consistent the higher geological data density therein. Thus, the surface grid is then extruded along the vertical direction to create layers whose thickness increases with depth according to the following criteria: … "

Added references

Carcano, C., Piccin. A.: Geologia degli acquiferi Padani della Regione Lombardia Regione Lombardia, Eni Divisione Agip, https://www.cartografia.regione.lombardia.it/metadata/acquiferi/doc/, 2001.

Regione Emilia-Romagna, ENI-AGIP, 1998. Riserve idriche sotterranee della Regione Emilia-Romagna. S.EL.CA, Firenze.

Regione Lombardia, ENI-AGIP, 2002. Geologia degli acquiferi padani della Regione Lombardia. S.EL.CA, Firenze.

Ricci Lucchi F, Colalongo ML, Cremonini G, Gasperi G, Iaccarino S, Papani G, et al. Evoluzione

sedimentaria e paleogeografica del margine appenninico (Sedimentary and palaeogeographic evolution of the Apenninic margin). Guida alla geologia del margine appenninico padano. Guide geologiche regionali, Soc. Geol. Ital.; 1982. p. 17–46.

**Comment #2:**

*In lines 265/266 a reference to the formulas where the targets of the calibration kc and rq might be added.*

**Answer #2:**

We thank the Reviewer for the comment. We will modify the manuscript as follows to improve clarity (see also answer to comment #6 of Reviewer #2).

"where, $\overline{h}_l$ and $h_l$ denote observed and estimated hydraulic head at well $l$, respectively. Estimation of permeability of each geomaterial ($k_c$ in Eq. 3) and of the correction coefficient ($r_q$ in Eq. 4) entails minimizing Eq. (6) (i.e., considering all available hydraulic head data, $N_{h_b}$)."

**Comment #3:**

*It is not clearly described whether the proportion of geomaterials, Fig. S1, is an a priori information or the result of calibration. As this is an important information anyway Fig. S1 should be included in the manuscript and not part of the supplementary material. In order to support the descriptions in l. 351 (and similar descriptions), it would be helpful to have a figure with the distribution of the geomaterials available.*

**Answer #3:**

We agree that including Figure S1 in the main body of the manuscript would enhance clarity. Additionally, a clear explanation of the origin of the information will be added in Section 3.2. We also add to the figure some information taken from the study of Manzoni et al. (2023) showing the geomaterial distribution upon which calculation of $f_{c,i}$ is grounded. The revised text now reads:

"Here, $N_i$ denotes the number of cells associated with the hydrostratigraphic model of Manzoni et al. (2023) that are included in the $i$-th cell of our simulation grid and $P_{c,j}$ is the probability that the $c$-th category (or geomaterial) be assigned to cell $j$ of the above mentioned hydrostratigraphic model. Figures 3a depicts the percentage of simulation grid cells associated with given (color-coded) ranges of values for $f_{c,i}$ cross each geomaterial category. Figure 3b illustrates the spatial distribution of the most probable geomaterial category within the Po River basin, as obtained by Manzoni et al. (2023). We then assess the permeability of the $i$-th cell of the numerical grid as

$$\overline{k}_i = \sum_c^{N_c} f_{c,i}\, k_c \qquad \text{with } N_c = 6 \qquad (3)$$

where $k_c$ is the permeability of the $c$-th category. Values of $k_c$ are estimated through model calibration, while $f_{c,i}$ is provided as prior information (see Manzoni et al., 2023). Details regarding model calibration are illustrated in Sect. 3.3."

[Figure]

(a)

(b)

**Figure 3: (a) Percentage of grid cells characterized by given ranges of values of $f_{c,i}$ (Eq. 2); (b) Spatial distribution of modal categories obtained by Manzoni et al. (2023). Planar maps are selected at 5, 10, 25, 50, 100, 150, 200, and 350 m below ground surface.**

…

"This finding is attributed to the fact that the simulation grid cells with the highest proportion of geomaterial five can be found in the mountainous areas and near the foothills (see Fig. 3b), which are close to the boundary where an inflow boundary condition is applied."

**Comment #4:**

*The concept behind the combination of the information in Fig. 4 is not really clear. Why are Fig. 4a and 4b combined with 4c and 4d? The information in Fig 4a might not really be important for the purpose of the manuscript. Fig 4b might be improved if the difference between simulate and observed heads are provided instead of the observed heads only. In Fig 4d the dark grey and dark red areas could not easily be distinguished as described in l. 359.*

**Answer #4:**

We understand the concern about the clarity of the combination of information in Figure 4. We will revise Figure 4 to improve readability and interpretation. Figure 4 will include a frequency distribution of the difference between simulated and observed heads, as suggested by the Reviewer. Furthermore, we will reorganize the material in the original Figure 4 into two figures, i.e., Figures 4 and 5. We will increase the color saturation of the image in Figure 4c and 4d to make the colors more distinguishable. The revised figures are included in the following.

[Figure]

**Figure 4: (a) Convergence analysis of $f_{N_b}$ and $f_{N_r}$ (Eq. 6); (b) observed versus simulated hydraulic heads (head values associated with the $N_{h_r}$ wells located close to the rivers are depicted in orange); (c) normalized frequency distribution of differences between observed and simulate hydraulic heads.**

[Figure]

**Figure 5: Covariance matrix of parameter estimates related to (a) Eq. (6) and (b) Eq. (7).**

**Comment #5:**

*The concept behind the combination of the information in Fig. 5 is not really clear. The description of Fig 5a, l. 372-380, is too coarse. The definition of the macro areas is not clearly motivated. There are more areas related to the several macro areas as described. In Fig 5b it does not become clear what the colour distribution in the represents. The comparison described in l. 367-369 does not really become clear from Fig. 5c.*

**Answer #5:**

We will include a general description of Figure 5 that elucidates the combination of the maps and graphs herein included. Additionally, we will revise the description in lines 372-380 of the original manuscript to provide a more detailed explanation. We will then ensure that the motivation behind the definition of these macro areas is clearly articulated to enhance readability.

To assist the Reviewer, we show below (Figure R.1) a graph of the frequency distribution of log-permeability values (vertical axis not in log scale, as opposed to the original Figure 5) to support the concept behind the identification of strongly homogeneous large volumes of the domain associated with the highest peaks in the frequency distribution of permeability values.

[Figure]

Figure R.1: Frequency distribution of natural logarithm of permeability, log($k$) ($k$ expressed in m$^2$).

We will revisit Figure 5c to enhance clarity of the comparison described in lines 367-369 of the original manuscript. In the following we add a proposal of revised text for the interest portion of the manuscript.

"Figure 5 offers an overview of the spatial distribution and vertical variation of permeability values ($k$) across the subsurface domain. Figure 5a depicts the frequency distribution of the estimated $k$ values. These results reveal three dominant modes (or peaks) in the distribution. These are characterized by a frequency that is one order of magnitude higher with respect to the rest of the $k$ values. This element suggests that the subsurface domain can be conceptualized as comprising three main macro-areas, each of these being characterized by (mostly) homogeneous spatial distributions of permeability values.

The spatial distribution of these macro area is consistent with the distribution of the three main sediment types indicated in the Italian Geological Map (Compagnoni et al., 2004) within the Po Plain (see Fig. 5c). Figure 5a provides an appraisal of the spatial distribution of the three macro-areas by means of envelopes obtained through projection of their otherwise three-dimensional shape onto a two-dimensional plane. This visualization is complemented by Fig. 5b, which depicts a qualitative representation of the vertical distribution of log($k$) along selected cross-sections (vertical exaggeration of 200). Access to a detailed grid of the three-dimensional distribution of $k$ is available through the code and data repository (https://doi.org/10.5281/zenodo.10697654).

The first macro-area, associated with the lowest permeability values within the modeled domain, generally corresponds to the south-eastern portion of the alluvial plain (Adriatic sector). Here, finer and less permeable sediments constitute the main features associated with geological deposition processes. The second macro-area is primarily located near the northern and western boundary, adjacent to the Alpine foothill areas, and is characterized by intermediate permeability values. Additional smaller areas with conditions similar to the Alpine foothills can be identified in the foothill areas of the Apennines. Note that, according to Éupolis Lombardia (2016), the planar area adjacent to the foothills in the Lombardia Region is very heterogeneous and features a series of highly permeable layers interspersed with less permeable layers. This is consistent with the intermediate range of permeability values obtained within our large-scale domain through model calibration. The third macro-area is characterized by high permeability values. It spans the entire depth of the system in the central-southern portion of the plain while it does not reach the surface in the northeastern part of the domain. This area is influenced by the deposits formed by the presence of the Po River.

[Figure]

(a)    (b)    (c)

**Comment #6:**

*Fig. 6 might be reorganized as different information, v and h, is combined. It does not become clear why different cross section are used in Fig 6 and Fig. 5.*

**Answer #6:**

We thank the Reviewer for the suggestion regarding the organization of Figure 6. We will reconsider the layout of the figure to improve the communication of the combined information on different variables. Additionally, we will modify Figure 5b (see answer to comment 4 above) to include the same vertical cross-sections used in Figure 6.

[Figure]

**Figure 1: Groundwater flow model outputs: (a) hydraulic head and distribution of groundwater fluxes across the top layer of the model; (b) magnitude and direction of groundwater flux and permeability distribution along cross-sections A-A', B-B', and C-C' (vertical exaggeration = 200).**

**Comment #7:**

*In order to emphasis the importance of the 3d approach it might be useful to describe the Morris indices for all model layers, especially the lower ones. If helpful an additional figure might be provided which might be added as supplementary material.*

**Answer #7:**

We agree that including a representation in a format that allows for a better understanding of the vertical distribution of Morris' indices can enhance the quality of the manuscript. We will add an accessible grid that contains 10 cross section into the data repository. We will modify the original manuscript to highlight the availability of these data in the open access data repository in Section 4.3 as follows.

"It is worth noting that all Morris indices display only modest variability along the vertical direction. The complete three-dimensional spatial distribution of $\mu^*_{\theta_p}$ and a grid containing 10 cross-sections highlighting our findings about the vertical variability of Morris indexes can be accessed in an open-source Visualization Toolkit (VTK) format for structured grids (Schroeder et al., 2006). These data are available in the code and data repository (https://doi.org/10.5281/zenodo.10697654)."

**Comment #8:**

*Fig. S2 might be included in the manuscript eventually in combination with Fig. 8 as this is an important result of the study.*

**Answer #8:**

Regarding the placement of Figure S2: We have opted to leave it in the supplementary material due to concerns about the paper density and length. We will of course abide by the Editor's decision on this matter.

**Comment #9:**

*A reference for the Penman-Monteith model should be added (l. 157) if this not covered by the reference 'Allen et al. (1998)'.*

**Answer #9:**

We will adjust the manuscript to enhance clarity as follows:

"For the evaluation of the actual evapotranspiration, *ET*, potential evapotranspiration is first computed by (*i*) making use of the model provided by Hargreaves and Samani (1985) in non-irrigated regions and (*ii*) combining the Penman-Monteith model with the correction crop coefficient in cultivated areas (consistent with Allen et al., 1998)."

technical corrections:

**Comment #10:**

*The sizes of the following figures should be increased. The names of the rivers should be readable in Fig. 1. The cross sections in Fig. 5b are not clearly visible. The sediment types in Fig. 5c are not clearly visible and a corresponding legend is missing. Details in the graphs in Fig. 6 are only hardly visible.*

**Answer #10:**

We will implement the suggestion of the Reviewer to improve quality of figures. We will increase the river name sizes in Figure 1. Vertical exaggeration of Figure 5b will be doubled and the prospective will be changed. We will increase the color exposure for Figure 5c. We will also modify Figure 6 according to comment 6 above, thus reducing redundancy and increasing image dimensions (see modified images at the answer of comments #5 and #6 above).

**Comment #11:**

*The formula '$Qs = rqR'sSs$' should treated as separate equation (l. 214). The further equations should be renumbered then.*

**Answer #11:**

We will implement the suggestion of the Reviewer to improve the clarity of the manuscript as follows.

$$Q_s = r_q R'_s S_s, \tag{4}$$

**Comment #12:**

*Within Fig 8 the number '6.5e5' is printed.*

**Answer #12:**

We will correct this oversight and apologize for the inconvenience.

---

## Author Comment (AC2)

POLITECNICO DI MILANO

**DEPARTMENT OF CIVIL AND ENVIRONMENTAL ENGINEERING**

Piazza Leonardo da Vinci, 32 I-20133, MILANO (Italy)

27th February 2024

**Re: Response to Comments of Reviewer #2-** *A Comprehensive Framework for Stochastic Calibration and Sensitivity Analysis of Large-Scale Groundwater Models* Submission to *Hydrology and Earth System Sciences*

We appreciate the efforts the Reviewer has invested in our manuscript. Following is an itemized list of the comments together with our response to each. Comments are listed in black italic font and our responses in blue font. Proposed revisions to the original text are in red fonts.

Sincerely,

Andrea Manzoni, Giovanni Michele Porta, Laura Guadagnini, Alberto Guadagnini, Monica Riva

*This paper presents a pioneering framework designed to model large-scale groundwater systems using a comprehensive three-dimensional approach. This approach not only accounts for the dynamics of river-aquifer interaction but also integrates three-dimensional spatial distributions of geo-materials. Notably, the framework excels in achieving a delicate equilibrium between the requisite simplification essential for large-scale groundwater models and the anticipated outcomes. Of particular significance is the utilization of a multi-objective optimization approach for model calibration and sensitivity analysis.*

We thank the Reviewer for the insightful review and positive feedback on our paper. We appreciate that you found our framework to be pioneering in its approach to modeling large-scale groundwater systems. The recognition of the value of our comprehensive three-dimensional approach, which considers both river-aquifer interaction dynamics and the spatial distributions of geo-materials, reinforces our determination in future research efforts in this context. We particularly appreciate the acknowledgment of the delicate equilibrium we sought to achieve between simplification and outcome accuracy in large-scale groundwater modeling. Striking this balance is indeed a key focus of our framework of analysis and modeling. We sincerely appreciate the time invested by the reviewer on our work and look forward to any further insights or suggestions.

From a general standpoint, I have two major concerns before recommending the paper publication:

**Comment #1:**

*This study, is build upon the substantial groundwork laid by Manzoni et al. (2023), a fact that merits explicit acknowledgment in both the introduction and the abstract. Indeed, in my opinion, the contributions made in Manzoni et al. (2023) represent a cornerstone of the framework presented herein, serving as more than just a dataset but rather as a fundamental component upon which this work is built.*

**Answer #1:**

We appreciate your recognition of the substantial groundwork laid by Manzoni et al. (2023), which indeed serves as a fundamental component of our framework. We agree that this element deserves explicit acknowledgment in both the introduction and the abstract, and we will revise these sections accordingly. Prompted by the Reviewer, we will cite Manzoni et al. (2023) more prominently throughout the paper (e.g., answer to comment #3 of Reviewer #1), highlighting their contributions to the field and how our work builds upon them. The revised manuscript will include the following paragraphs.

Into the Abstract:

"We address the challenges posed by the characterization of the heterogeneous spatial distribution of subsurface attributes across large-scale three-dimensional domains upon incorporating a recent probabilistic hydrogeological reconstruction specific to the study case."

Into the introduction:

"In some cases, system properties are assumed to be constant along the vertical direction (e.g., Maxwell et al., 2015; Shrestha et al., 2014; Soltani et al., 2022) without taking into account the three-dimensional nature of the spatial heterogeneity of the subsurface system. In this sense, parameter values are typically inferred from literature information (Naz et al., 2023; Maxwell et al., 2015), thus possibly introducing large margins of uncertainty that are seldom quantifiable. The work of Manzoni et al. (2023) addresses these challenges by proposing a machine-learning-based methodology for delineating the spatial distribution of geomaterials across large-scale three-dimensional subsurface systems. These authors showcase their approach upon focusing on the Po River Basin in northern Italy. Their work provides a comprehensive dataset comprising lithostratigraphic data from various sources and offers a robust framework for quantifying prediction uncertainty at each spatial location within the reconstructed domain. Hence, our study rests on the findings of Manzoni et al. (2023). In these sense, the latter serve as more than simply a dataset but as a critical component upon which we build our groundwater flow model calibration.

…

The approach involves the development of a groundwater model that includes a probabilistic three-dimensional hydrogeological reconstruction of the investigated area. As stated above, we do so upon integrating the result illustrated by Manzoni et al. (2023). Specifically, we leverage on their probabilistic three-dimensional hydrogeological reconstruction, which enables us to infer the spatial distribution of geological properties at a scale that was previously unattainable. By incorporating this

advanced hydrogeological reconstruction into our workflow, we address key challenges posed by uncertainties that are inherent to large-scale groundwater systems."

**Comment #2:**

*The aspects presented in lines 59-61 may not inherently appear novel 'per se'. The authors should thus better emphasize the unique contributions and novelties of their work to distinguish it from existing literature.*

**Answer #2:**

Prompted by the Reviewer, we will explicitly state the novel aspect of our approach right after lines 59-61 of the original manuscript. This will involve clarifying how our work builds upon (or departs from) existing research in a way that contributes to yield new insights or solutions. We will reassess the wording and framing of the paragraph pointed out by the Reviewer to ensure that it effectively communicates the original features of our study. The revised text now reads:

"Here, we introduce and test a methodological approach for stochastic model calibration tailored to large-scale scenarios (exceeding 10,000 km²). Our proposed methodology combines a suite of tools that have not been previously employed to address groundwater modeling at such a vast scale and with such level of system complexity. This includes (*a*) modeling the dynamics of groundwater movement across a three-dimensional setting, (*b*) embedding and analyzing in details interactions between rivers and aquifers, (*c*) relying on a probabilistic reconstruction of geological material distributions and attributes, (*d*) resting on multi-objective optimization techniques for stochastic calibration of large-scale groundwater models, and (*e*) performing a detailed informed global sensitivity analysis to assess the degree of spatial variability of the relative importance of uncertain model parameters therein. Through incorporation of these tools, our methodological and operational workflow yields a calibrated model that enhances understanding of aquifer dynamics from a holistic perspective and reveals insights into the spatial pattern of the sensitivity of model outputs to model parameters. Results associated with the latter element can be employed to inform future data acquisition efforts to improve model parameterization and hydraulic head estimates and emphasize the need to balance model complexity with simplifications that might be required to tackle large-scale groundwater modeling."

Other specific comments:

**Comment #3:**

*In light of Manzoni et al. (2023), the statement highlighting the limitations of data-driven models due to constraints in the quantity and quality of available training data, particularly in large-scale scenarios where data accessibility across the entire domain might be limited, warrants revision. Specifically, it should be contextualized to reflect insights gleaned from Manzoni et al. (2023) and potentially revised to underscore the advancements or strategies proposed in this study to address such challenges.*

**Answer #3:**

We appreciate the Reviewer's insightful comment. The constraint on data availability currently pertains specifically to groundwater flow and pressure data, at least in the area analyzed. As outlined in Section 3.3.1 (Data curation), our study relies on a limited dataset comprising 286 wells where data can be employed for groundwater model calibration. The study by Manzoni et al. (2023) is focused on the reconstruction of the spatial distribution of geomaterials and leverages on a robust dataset associated with more than 50,000 wells. This yields more than 2 million data points associated with geological information and adequately supports the application of data-driven machine learning approaches for hydrogeological reconstruction. It is also noted that localized areas exhibiting high uncertainty in the geological material distribution due to limited data availability primarily lie outside the lateral boundary of the modeled groundwater system (see also Manzoni et al. (2023)). The revised manuscript will include the following two modifications and additions in Introduction and Section 3.2., respectively.

"It is noted that data-driven models are heavily constrained by the quantity and quality of available training data. In this context, groundwater flow and pressure data may not be as readily accessible as, for example, lithostratigraphic data (see e.g., Manzoni et al., 2023) across the entire domain, especially when considering large-scale scenarios.

…

Notably, the highest uncertainties in the hydrostratigraphic reconstruction model are found beyond the lateral boundaries of the groundwater system, where only a limited number of investigations is available."

**Comment #4:**

*The clarity of Equation "$Qs = rqRsSs$" is ambiguous. Assuming my interpretation is correct, where the subscript $s$ represents the $s$-th subbasin, further clarification from the author is needed regarding whether the efficiency term is constant across time or space. Additionally, a more detailed discussion on the assumption of the correction coefficient, $r\_q$, being constant for all subbasins would enhance understanding. This could include explanations of the rationale behind this assumption, any potential implications or limitations, and how it aligns with existing literature or empirical evidence.*

**Answer #4:**

Prompted also by a comment of Reviewer #1, the expression "$Qs = rqRsSs$" is now addressed as a standalone equation within the manuscript. This adjustment enables us to provide a comprehensive elucidation of the underlying assumptions guiding its formulation. Introducing individual $r_q$ values for each sub-basin would increase the level of complexity of the parameterization process, potentially resulting in model over-parameterization and subsequent overfitting of data. Notably, Figure 7 documents that simulated hydraulic heads demonstrate increased sensitivity to $r_q$ near the lateral boundaries of the domain. In this context, fine-tuning $r_q$ values for each sub-basin could lead to overfitting. We will implement these considerations in Section 3.2 of the revised manuscript. The revised text now reads:

"Making use of the results of Sect. 3.1, we evaluate within each of these sub-basins the average (in time) amount of water that infiltrates within a day as

$$Q_s = r_q R'_s S_s, \tag{4}$$

where $R'_s$ [L/T] is the (space-time averaged) recharge rate evaluated for the $s$-th sub-basin (with ground surface area of $S_s$) during the temporal window spanning the years 2013-2019. To account for possible exfiltration of infiltrated water or infiltration of water due to surface-groundwater interaction (e.g., river water infiltration), we also introduce a correction coefficient, $r_q$. The latter is set at a constant value for all sub-basins to avoid model overparameterization and is estimated through model calibration, as detailed in Sect. 3.3.

**Comment #5:**

*The description of the domestic water flux associated with groundwater resource utilization lacks the necessary details for reproducibility and applicability in other areas. To enhance the transparency and replicability of the work, it is suggested that the authors provide additional information regarding the data utilized, including its sources and any assumptions or hypotheses underlying its selection or interpretation. This could involve detailing the methodology for acquiring the total volumetric flow rate associated with groundwater extractions of drinking water within a municipality, as well as specifying the criteria used to define the surface area covered by the municipality. By providing this supplementary information, the study's findings can be better contextualized and applied to different geographical regions or scenarios.*

**Answer #5:**

In response to the Reviewer's suggestions, we will detail the methodology used to acquire the total volumetric flow rate associated with groundwater extractions for drinking water within each municipality. Furthermore, we will specify the criteria used to define the surface area covered by each municipality, ensuring clarity and consistency in our spatial delineation. In this context, we will modify the text in the designated section as follows.

"To estimate the volumetric flow rate for domestic use, we rely on the public water supply data provided by the Italian National Institute of Statistics (ISTAT, 2020). This dataset contains values of flow rates (in m$^3$/year) employed for domestic purposes for each municipal administrative area and the share of domestic water associated with groundwater resources. Such data are available for the years 2012, 2015, 2018, and 2020. We then evaluate the average flow rate for each municipality on these bases. For a given municipality, domestic water fluxes associated with the use of groundwater resources is assessed upon evaluating the ratio of the total volumetric flow rate associated with groundwater extractions for drinking water to the surface area covered by the municipality itself (OpenStreetMap, 2021). Volumetric flow rates employed in the model are then obtained by multiplying the portion of the municipality area within the modeled domain by the domestic water flux. Due to the lack of comprehensive information regarding the location of extraction wells, we consider such a water flow rate as a distributed sink term located within the deepest layer of the simulation domain beneath each associated municipality. This assumption is grounded on the notion that drinking water wells are typically engineered to extract water from locations that are protected from potential contaminants that may infiltrate and pollute shallower regions of subsurface water bodies."

**References**

OpenStreetMap. OpenStreetMap database [PostgreSQL]. OpenStreetMap Foundation: Cambridge, UK; 2021. https://gisdata.mapog.com/italy/Municipality%20level%204

**Comment #6:**

*The section pertaining to multi-objective calibration appears to lack clarity and would benefit from a more precise formulation of the mathematical framework. It is recommended to clearly define the two objective functions, delineate the independent variables, and specify any constraints involved in the optimization process. Providing explicit details about the mathematical formulation will enhance understanding and facilitate the replication of the calibration methodology.*

**Answer #6:**

In response to this suggestions, we will revise the section to explicitly define the two objective functions separately. We move the algorithm description from Appendix A into the text and improve the algorithm description upon focusing on mathematical formulation details to assist replicability of the calibration methodology (see also answer to comment #7). To further improve the description and the replicability of the implemented algorithm, we also plan to add an image containing the pseudocode. The revised manuscript now reads:

"Model parameters are estimated through a multi-objective optimization approach. The latter is tied to the joint minimization of two objective functions formulated as

$$f_{N_{h_b}} = \sqrt{\frac{\sum_{l=1}^{N_{h_b}}(\overline{h_l} - h_l)^2}{N_{h_b}}} \tag{6}$$

and

$$f_{N_{h_r}} = \sqrt{\frac{\sum_{l=1}^{N_{h_r}}(\overline{h_l} - h_l)^2}{N_{h_r}}} \tag{7}$$

where $\overline{h}_l$ and $h_l$ denote observed and estimated hydraulic head at well $l$, respectively. Estimation of permeability of each geomaterial (i.e., $k_c$ in Eq. 3) and of the correction coefficient (i.e., $r_q$ in Eq. 4) entails minimizing Eq. (6) (considering all available hydraulic head data, $N_{h_b}$). To estimate the specific conductance of the riverbeds, $\alpha_r$ (with $r = 1, \dots, 20$), we minimize Eq. (7) with $N_{h_r} < N_{h_b}$, where $N_{h_r}$ is the number of wells located within a maximum distance of 5 km from a river (see orange dots in Fig. 1). Including this constraint on the distance between a river and observation wells enables us to refine the estimation of $\alpha_r$ by considering only hydraulic head observations that are significantly impacted by the interconnection between the groundwater system and the rivers. Note that minimization of Eq. (6) and (7) is tantamount to relying on a Maximum Likelihood (ML) estimation approach assuming that measurement errors of hydraulic head are not correlated and can be described through a Gaussian distribution (Carrera and Neuman, 1986)."

…

The implemented algorithm is designed to address global optimization problems through alternate evolution of candidate solutions between the two different species. The algorithm uses mutation, crossover, and selection strategies to enhance the quality of solutions as detailed in the following. First, we introduce the populations of candidate solutions. For each of the two species (where 'sp' takes the values of one or two, for species associated with Eq. (6) or (7), respectively), we consider a set of $N_S$ candidate solutions (or members), denoted as $\mathbf{S_{sp}} = [\mathbf{s}_{sp,1}, \dots, \mathbf{s}_{sp,m}, \dots, \mathbf{s}_{sp,N_s}]$. Following Storn and Price (1997), we set $N_S = 15 \times N_p$, $N_p$ being the number of parameters (i.e., $N_p = 7$ or 20 for Eq. 6 or Eq. 7, respectively). Initial candidate solutions are defined by randomly selecting parameter values from a parameter space whose extent is designed to encompass a broad range of possible solutions. Members of the populations are combined and mutated to calculate the next generations of candidate solutions as follows. We start by computing a mutated vector for each $m$-th candidate solution of a species associated with the $k$-th iteration of the optimization algorithm (or generation) as:

$$\hat{\mathbf{s}}_{sp,m}^k = \mathbf{s}_{sp,m}^k + F\left(\mathbf{s}_{sp,a}^k - \mathbf{s}_{sp,b}^k\right), \tag{8}$$

Here, $F$ represents an algorithm parameter (termed differential weight) that is set equal to 0.5 and $\mathbf{s}_{sp,a}^k$ and $\mathbf{s}_{sp,b}^k$ (with $a \neq b \neq m$) correspond to two (randomly selected) members of the population. We then combine parameters of $\hat{\mathbf{s}}_{sp,m}^k$ and $\mathbf{s}_{sp,m}^k$ to determine the trial vector $\tilde{\mathbf{s}}_{sp,m}^k$: if a parameter of $\tilde{\mathbf{s}}_{sp,m}^k$ is selected for mutation, its value is taken from $\hat{\mathbf{s}}_{sp,m}^k$; otherwise, it is taken from $\mathbf{s}_{sp,m}^k$. We randomly choose the parameters of $\mathbf{s}_{sp,m}^k$ that will undergo mutation among the parameters associated with the $sp$ species, with a probability of parameter mutation set to 0.5. We finally select the $m$-th candidate solution of the $(k+1)$-th generation, $\mathbf{s}_{sp,m}^{k+1}$, by comparing the trial member, $\tilde{\mathbf{s}}_{sp,m}^k$, and the $m$-th population member from the $k$-th generation, $\mathbf{s}_{sp,m}^k$, based on the following condition:

$$\mathbf{s}_{sp,m}^{k+1} = \begin{cases} \tilde{\mathbf{s}}_{sp,m}^k, & if\, f_N\left(\tilde{\mathbf{s}}_{sp,m}^k\right) < f_N\left(\mathbf{s}_{sp,m}^k\right) \\ \mathbf{s}_{sp,m}^k, & if\, f_N\left(\tilde{\mathbf{s}}_{sp,m}^k\right) \geq f_N\left(\mathbf{s}_{sp,m}^k\right) \end{cases} \qquad \text{with } f_N = f_{N_{h_l}} \text{ or } f_{N_{h_r}}. \tag{9}$$

The algorithm steps can be summarized as follows at a given iteration $k$:

1. Calculate a new generation $(k+1)$ of the first species using Eq.s (8)-(9) with $f_N = f_{N_{h_l}}$, while keeping the parameters of the second species fixed;

2. Transfer the parameter set with the best performance, $\mathbf{s}_{1,best}^{k+1}$, among the members of $\mathbf{s}_{1,m}^{k+1}$ to the second species;

3. Maintain the parameters of the first species as fixed while calculating $\mathbf{s}_{2,m}^{k+1}$ (the next generation of the second species), thus repeating step 1 for the second species with $sp = 2$ and Eq. (7);

4. Pass back to the first species the parameter set of the member in the second species with the best objective function value, $\mathbf{s}_{2,best}^{k+1}$;

5. repeat steps 1 to 4 until a stopping criterion is met.

The patience stopping criterion is here employed for both objective functions, i.e., the algorithm stops if no improvement in performance over 80 consecutive iterations (or epochs) is detected. Figure 3 illustrates the Pseudocode of the algorithm.

```
Begin
    Initialize $S_1 = \left[s_{1,1}, \ldots, s_{1,m}, \ldots, s_{1,N_{s1}}\right]$ and $S_2 = \left[s_{2,1}, \ldots, s_{2,m}, \ldots, s_{2,N_{s2}}\right]$.
    Evaluate the members of $S_1$ using Eq. (6).
    Evaluate the members of $S_2$ using Eq. (7).
    While stopping criteria are not met for both the species:
        For each $m$-th member of $S_1$
            Create $\tilde{s}_{1,m}^k$ by applying mutation and crossover.
            Evaluate $\tilde{s}_{1,m}^k$ using Eq. (6).
            Select $s_{1,m}^{k+1}$ according to Eq. (9).
            Select $s_{1,best}^{k+1}$
        EndFor
        Fix the parameters associated with Eq. (6) equal to $s_{1,best}^{k+1}$.
        For each $m$-th member of $S_2$
            Create $\tilde{s}_{2,m}^k$ by applying mutation and crossover.
            Evaluate $\tilde{s}_{2,m}^k$ using Eq. (7).
            Select $s_{2,m}^{k+1}$ according to Eq. (9).
            Select $s_{2,best}^{k+1}$
        EndFor
        Fix the parameters associated with Eq. (7) equal to $s_{2,best}^{k+1}$.
        k++
    EndWhile
    Return $s_{2,best}^{k+1}$ and $s_{1,best}^{k+1}$.
End
```

Figure 3: Pseudocode of the employed algorithm.

**Comment #7:**

*The authors should provide further clarification regarding the statement "We note that CCDE does not require defining a single weighted multi-objective function, as otherwise required by the standard DE." It would be beneficial to elaborate on the distinction between the adopted algorithm (CCDE) and the standard NSGA-II algorithm. For example, the NSGA-II algorithm does not inherently require a weighted multi-objective function either. Algorithms such as NSGA-II typically operate based on the concept of dominance and the Pareto front without necessitating explicit weight assignment during the optimization process. Therefore, expanding on the characteristics of CCDE and how it differs from conventional multi-objective optimization algorithms like NSGA-II would provide valuable insights. Furthermore, it appears that the original paper by Storn and Price (1997) primarily focuses on single-objective optimization, similar to the work by Trunfio in 2015. Given that the current study deals with non-contrasting objective functions, it may explain the absence of weight utilization. However, providing additional details on the rationale behind the choice of CCDE and its suitability for handling the specific characteristics of the optimization problem in this context would enhance understanding.*

**Answer #7:**

As the Reviewer correctly points out, the Coevolutionary Differential Evolution (CCDE) proposed by Trunfio (2015) is designed for single-objective optimization. In our case, the coevolutionary algorithm is adapted to handle two objective functions. Specifically, the two evolving species are associated with two different objective functions (i.e., Eq. 6 and Eq. 7 in the revised manuscript). In this context, we revise the manuscript according to our answer to comment #6 to guarantee replicability of the methodology.

We plan to revise the sentence highlighted by the Reviewer to clarify that we have implemented a modified version of the coevolutionary differential evolution algorithm. We also intend to amend the original manuscript to underscore the main differences and similarities between the algorithm we used and the NSGA II as follows.

"The two objective functions to minimize are closely interconnected. We implement an enhanced variant of the Differential Evolution (DE) optimization method (Storn and Price, 1997) to effectively minimize both objective functions simultaneously. Here, we rest on a modified version of the Cooperative Coevolutionary Differential Evolution (CCDE) optimization algorithm proposed by Trunfio (2015) tuned to our problem, which is an adaptation of the standard DE algorithm to the theory of Coevolutionary Algorithms (CAs). The implemented algorithm does not require defining a single weighted multi-objective function, as otherwise required by standard DE and standard CCDE. Thus, our approach eliminates the non-trivial task of determining the appropriate (relative) weights between each of the terms that constitute the multi-objective function (e.g., Dell'Oca et al., 2023). Resorting to a modified CCDE algorithm enables us to balance between simplicity and the efficiency documented for CAs when dealing with multi-objective fitness functions (Khan et al., 2022).

As nature-inspired optimization techniques, CAs draw upon principles of biological coevolution, where optimization problems are linked to coevolving species (Dagdia and Mirchev, 2020). CAs share similarities with Evolutionary algorithms, as their sampling mechanisms and dynamics are inspired by Darwin's Theory of Evolution. Just as species evolve based on their fitness to survive and reproduce within an environment, solutions within a search space evolve to achieve the minimum of an objective function (Simoncini and Zhang, 2019). Additionally, the coevolution principle considers that a change in one species can trigger changes in related species, thus leading to adaptive changes in each species (Khan et al., 2022). In this context, Eq.s (6) and (7) represent optimization functions for two coevolving species. These are then optimized through the modified CCDE. Our algorithm differs from CCDE (Trunfio, 2015) primarily in the way we define the dimensions of the two species. Instead of employing random or dynamic grouping strategies (Zhenyu et al., 2008; Trunfio, 2015), we opt for a supervised grouping strategy linking one of the model parameters (i.e., riverbed conductance, $\alpha_r$) to one species and the remaining parameters to the other species.

We choose a modified version of Coevolutionary Differential Evolution (CCDE) over the widely used NSGA II (or its variant CC-NSGA-II) for our algorithm. Both these algorithms use a divide-and-conquer strategy and are effective for high dimensional optimization. However, while NSGA II relies on a genetic algorithm, our algorithm utilizes Differential Evolution (DE). According to Tusar and Filipic (2007), DE-based algorithms outperform GA-based algorithms in multi-objective optimization

due to a more efficient exploration of the parameter space. This element is particularly critical when optimal solutions lie on parameter bounds or amidst many local optima.

Additionally, our implemented algorithm does not explicitly optimize a *front*, which is otherwise a central concept in NSGA-II. Instead, it focuses on optimizing individual objective function values without introducing a dominance concept considering both objectives. This approach leads to a single set of optimized parameters, thus simplifying the optimization process through a balance of the contribution of both objective functions."

**References**

Yang, Z., Tang, K., and Yao X.: Large scale evolutionary optimization using cooperative coevolution, Information Sciences, 178, 2985-2999, https://doi.org/10.1016/j.ins.2008.02.017, 2008.

Tusar, T., and Filipic, B.: Differential evolution versus genetic algorithms in multiobjective optimization, Evolutionary Multi-Criterion Optimization, Springer Berlin Heidelberg, 257-271, https://doi.org/10.1007/978-3-540-70928-2_22, 2007.

**Comment #8:**

*It is suggested that the authors include a figure illustrating the framework chain, depicting the data inputs required as well as the expected outcomes. This visual representation will aid in comprehending the workflow of the methodology and provide a clear overview of the research process. Additionally, the figure can serve as a useful reference for readers to understand how various components of the framework interact and contribute to the overall analysis.*

**Answer #8:**

Prompted by the Reviewer, we will include the following new figure illustrating the framework chain to enhance clarity.

[Figure]

**Figure 1: Conceptual workflow for stochastic calibration and sensitivity analysis of large-scale groundwater models.**